# DCPO: DYNAMIC CLIPPING POLICY OPTIMIZATION

## ABSTRACT

Reinforcement Learning from Verifiable Rewards (RLVR) has emerged as a promising framework for enhancing the reasoning capabilities of large language models. However, existing approaches, such as GRPO, often suffer from zero gradients. This problem mainly stems from (i) fixed clipping bounds for token-level probability ratios and (ii) the standardization of identical rewards, which can lead to ineffective gradient updates and underutilization of generated responses. In this work, we propose **D**ynamic **C**lipping **P**olicy **O**ptimization (**DCPO**). DCPO (i) introduces a dynamic clipping strategy that adaptively adjusts clipping bounds based on token-specific prior probabilities to enhance token-level exploration, and (ii) employs a smooth advantage standardization technique that standardizes rewards across cumulative training steps to improve the response-level effective utilization of generated responses. DCPO achieved state-of-the-art performance on four benchmarks based on four different models. Specifically, on the AIME-24 benchmark, DCPO reaches an Avg@1 of 46.7 (greedy decoding) and an Avg@32 of 38.8 (32-sample decoding) with the Qwen2.5-Math-7B model, surpassing DAPO (36.7/31.6), GRPO (36.7/32.1) and GSPO (40.0/34.9). On the AIME25 benchmark based on Qwen2.5-14B, DCPO achieves a performance of (23.3/19.0), surpassing GRPO (13.3/10.5), DAPO (20.0/15.3) and GSPO (16.7/9.9). Furthermore, DCPO achieved an average 28% improvement in the nonzero advantage over GRPO in four models, doubled the training efficiency compared to DAPO, and significantly reduced the token clipping ratio by an order of magnitude compared to both GRPO and DAPO, while achieving superior performance. These results demonstrate DCPO's effectiveness in leveraging generated data more efficiently for reinforcement learning in large language models.

## 1 INTRODUCTION

In recent years, the reasoning capabilities of large language models (LLMs) have garnered increasing attention (OpenAI, 2025a; Guo et al., 2025; Comanici et al., 2025). Reinforcement Learning from Verifiable Rewards (RLVR) represents a promising technique for enhancing reasoning capabilities. This method leverages rule-based outcome rewards to optimize LLMs, specifically enhancing the performance of LLMs in domains requiring reflective decision-making, such as mathematics and coding (DeepSeek-AI, 2024).

To support RLVR, the GRPO (DeepSeek-AI, 2024; Liu et al., 2025; Yuan et al., 2025; Zheng et al., 2025) family of algorithms was developed. Although the robust performance of GRPO has been extensively validated in numerous studies, recent research, such as DAPO (Yu et al., 2025), has identified critical limitations. Specifically, the GRPO suffers from entropy collapse, characterized by the advantage values decaying to zero during later training stages. Furthermore, inappropriate weight-clipping can discard rarely sampled learning signals, thus overly limiting the diversity of model exploration.

To address these challenges, for example, DAPO introduces the Clip-Higher strategy to mitigate entropy collapse by relaxing the upper clipping bound, and the Dynamic Sampling strategy to filter out samples with uniform rewards, ensuring that every batch contains only samples with valid gradients. However, these methods have two notable limitations: (i) **Response-Level Inefficiency**: Dynamic sampling inherently reduces the sampling efficiency, leading to slower training convergence. (ii) **token clipping deficit**: Our analysis reveals that the optimal clipping bound varies across tokens, making

a uniform upper bound suboptimal. Although some others avoid directly discarding zero-gradient responses, these limitations remain prevalent.

In this work, we introduce **D**ynamic **C**lipping **P**olicy **O**ptimization (**DCPO**), a novel approach designed to address the key limitations of previous methods. First, we propose a **dynamic clipping method that adaptively adjusts the clipping bounds based on the old token-specific probabilities** to mitigate the drawbacks of the fixed clipping bounds, rather than setting different fixed lower and upper clipping bounds. Specifically, tokens with lower prior probabilities are granted wider clipping bounds, allowing more space for token-level exploration(see Section 2 and Appendix A.6). Second, we introduce a **smoothing technique that standardizes advantages across cumulative steps** to address the gradient descent problem caused by zeroing out the advantages of responses with the same reward. Our approach aggregates the reward distribution of all cumulatively generated responses with the current generated distribution, allowing more efficient response-level exploration(see Section 3). In addition, DCPO only computes the mean loss across tokens within each individual response instead of averaging over an entire batch, which preserves the relative advantage structure among responses to the same prompt and avoids the dilution effect of batch-level averaging.

Finally, we merged Math-DAPO-17k (Yu et al., 2025) with levels 3-5 of the MATH dataset (Liu et al., 2025) as the training dataset and selected four common mathematical reasoning benchmarks to evaluate performance. We conducted experiments on four different models and used GRPO, DAPO and (where available) GSPO as baselines. The experimental results show that DCPO consistently achieved the superior preference, with particularly large gains on the challenging AIME benchmarks: on AIME24-Avg@32 with 32-time sampling, DCPO-7B scored 38.8 vs. 32.1 (GRPO) vs. 31.6 (DAPO) vs. 34.9 (GSPO), and on AIME25-Avg@32, DCPO-14B reached 19.0 vs. 10.5 (GRPO) vs. 15.3 (DAPO) vs. 9.9 (GSPO). Moreover, DCPO increased the average nonzero advantage ratio by 28% over GRPO, doubled training efficiency over DAPO, and reduced token clipping by an order of magnitude compared to GRPO and DAPO, while delivering superior performance.

For preliminary details, including loss and advantage calculation methods for previous algorithms such as GRPO, DAPO and GSPO, see the Appendix A.3.

## 2 DYNAMIC-ADAPTIVE CLIPPING BOUNDS FOR RL(**DAC**)

Clipping the probability ratio is a widely adopted technique in reinforcement learning to stabilize policy updates and prevent large, destabilizing parameter shifts during optimization. This mechanism is particularly important when applying importance sampling in off-policy settings. When importance sampling is reweighted by the ratio $r(x)$ between the new probability $p(x)$ and the old probability $q(x)$, an unbiased estimate of $\mathbb{E}_{q(x)}[f(x)\frac{p(x)}{q(x)}]$ can be obtained, but the variance of such an estimate can be significantly inflated, as described in Equation (1).

$$\text{Var}_{x \sim q}\left[f(x)\frac{p(x)}{q(x)}\right] - \text{Var}_{x \sim p}\left[f(x)\right] = \mathbb{E}_{x \sim p}\left[f(x)^2(\frac{p(x)}{q(x)} - 1)\right] = \int f(x)^2(\frac{p(x)}{q(x)} - 1)p(x)\text{d}x \tag{1}$$

To mitigate this variance inflation, prior works(e.g., Schulman et al. (2017)) constrain the probability ratio within a fixed clipping bound defined by the hyperparameter $\epsilon$ as Equation (2).

$$|r(x) - 1| \leq \epsilon \tag{2}$$

However, a symmetric fixed clipping bound constrains the model's exploration of the diversity space, particularly in regions where the old policy assigns low probability mass. In these scenarios, the absolute bound for decreasing probabilities is inherently limited, restricting the capacity for meaningful policy updates. Although methods like DAPO attempt to address this issue with asymmetric clipping bounds, the fundamental limitation persists. Furthermore, GSPO discards entire responses when the sequence-level importance ratio exceeds the asymmetric clipping bounds, wasting more informative tokens, and retaining more high-variance tokens leading to training instability.

From the perspective of controlling variance bias, a more practical and reasonable approach is to impose probability-dependent constraints, as formalized in Equation (3).

$$|(r(x) - 1)p(x)| \leq \epsilon \tag{3}$$

Substituting $p(x) = r(x)q(x)$ into Equation (3) and ensuring non-negativity of the square root term by taking its maximum with zero, we obtain a closed-form solution for $r(x)$. This yields **dynamic-adaptive clipping bounds** that vary adaptively with $q(x)$ and hyperparameters $\epsilon_{low}$ and $\epsilon_{high}$, as specified in Equation (4).

$$0.5 + \frac{1}{2}\sqrt{\max\left(1 - \frac{4\epsilon_{low}}{q(x)}, \; 0\right)} \leq r(x) \leq 0.5 + \frac{1}{2}\sqrt{1 + \frac{4\epsilon_{high}}{q(x)}} \tag{4}$$

This dynamic-adaptive clipping method allows for greater policy exploration in low-probability regions. Furthermore, inspired by the dual clipping method (Ye et al., 2020), we set maximum bounds $r_{max}$ for positive and negative advantages to prevent excessive clipping bounds from overtraining. For detailed derivation and analysis of these dynamic clipping bounds from the importance sampling framework, please see Appendix A.5 and Appendix A.6.

## 3 SMOOTH ADVANTAGE STANDARDIZATION(**SAS**)

In previous works, such as GRPO, DAPO and GSPO, the advantage $\hat{A}_{new,j}^i$ of the $j$-th response is the standardized result of only taking the rewards $R_{j=1,\ldots,G}^i$ for the same prompt at the $i$-th step, as Equation (11) in the Appendix A.3.1. However, this approach can lead to several issues: **(i)** when randomness in response sampling causes all rewards to be the same in a given step, the advantage becomes zero, preventing the prompt from contributing to parameter updates despite potentially valuable differences in reasoning trajectories. **(ii)** randomness in high-entropy sampling can yield highly skewed label counts, causing large fluctuations in standardized advantage values across steps, even reversing signs, thus destabilizing training.

At the same time, recent work (Yue et al., 2025) suggests that RLVR primarily refines a policy to adjust the likelihood of correct samples rather than fundamentally altering the existing distribution. Given these considerations, the overall reward distribution of responses to the same prompt can be considered as drawn from the same global distribution throughout the course of training, which motivates a *cumulative* standardization as Equation (5). We can obtain cumulative statistics only using the statistics of the latest and current step without storing all previous responses. The additional memory space required is much smaller than the size of the training dataset (see Appendix A.7).

$$\hat{A}_{total,j}^i = \frac{\left(R_j^i - \mu_{total}^i\right)}{\sigma_{total}^i} \tag{5}$$

where the statistics are computed over all responses generated so far for the same prompt.

To mitigate fluctuations in step-specific standardization $\hat{A}_{new,j}^i$ and cumulative standardization $\hat{A}_{total,j}^i$, we introduce two smoothing functions, $\hat{SA}_{new,j}^i$ and $\hat{SA}_{total,j}^i$, which represent the weighted average between the two standardization methods with weight change over step $i$ in Equation (6).

$$\hat{SA}_{new,j}^i = \frac{i-1}{i}\hat{A}_{new,j}^i + \frac{1}{i}\hat{A}_{total,j}^i, \; \hat{SA}_{total,j}^i = \frac{1}{i}\hat{A}_{new,j}^i + \frac{i-1}{i}\hat{A}_{total,j}^i \tag{6}$$

$\hat{SA}_{new,j}^i$ places more weight on the current distribution as training progresses, reflecting a preference for adapting to the latest policy outputs. Conversely, $\hat{SA}_{total,j}^i$ places more weight on the overall cumulative distribution, viewing the deviation of the current step as a fine-tuning adjustment within the broader distribution. Finally, our final advantage $\hat{A}_j^i$ is defined as the smoothed advantage with the smaller absolute value to reduce the impact of the respective fluctuations of cumulative and current-step standardization on training stability, defined as Equation (7).

$$\hat{A}_j^i = \begin{cases} \hat{SA}_{new,j}^i, & \text{when } |\hat{SA}_{new,j}^i| < |\hat{SA}_{total,j}^i| \\ \hat{SA}_{total,j}^i, & \text{otherwise} \end{cases} \tag{7}$$

Once a prompt participates in the model optimization, its responses generated in the subsequent steps will participate in the model optimization regardless of whether the current advantage is 0 or not. Consequently, when identical rewards occur within a step, these responses will also update the model with the advantage of $\frac{1}{i}\hat{A}^i_{total,j}$, preserving useful learning signals and improving data efficiency.

## 3.1 OTM LOSS

Although the **T**oken-**L**evel **M**ean loss(**TLM**) (the loss Equation (13) used in DAPO) balances all token contributions, it disrupts the relative response-level relationship between different responses of different lengths. For instance, consider two responses to the same prompt: The response $A$ has an advantage of 1 and a length of 500, and the response $B$ has an advantage of 0.5 and a length of 1500. Under TLM, the loss for $A$ is weighted by $\frac{500}{500+1500} = 0.25$, and for $B$ by $\frac{1500}{500+1500} = 0.75$. Therefore, the total loss contribution of $A$ is $1 \times 0.25 = 0.25$ and of $B$ is $0.5 \times 0.75 = 0.375$. This results in $B$ having a larger influence (0.375) despite its lower advantage (0.5).

At the same time, for the average across batches, the **S**equence-**L**evel **M**ean loss (**SLM**) (the loss Equation (12) used in GRPO) weakens the size of the advantage, equivalent to reducing it by $G$ times after calculation for advantages. Especially when the batch size is large, it will destroy the standardized results of the original advantages between the correlations. We believe that standardized advantages already have relative relationships across responses, and the average across all tokens in the same response using $|o_i|$ not only maintains the original relative relationship, but also distributes the importance equally to each token within the response. Therefore, we only averaged the tokens within a given response without averaging all responses in the same batch. This is is referred to as the **O**nly **T**oken **M**ean loss (**OTM**) as Equation (8).

$$\mathcal{T}_{\text{DCPO}}\left(\theta\right) = \sum_{i=1}^{G} \frac{1}{|o_i|} \sum_{t=1}^{|o_i|} \min\left(r_{i,t}(\theta)\hat{A}_{i,t}, \ \text{clip}\left(r_{i,t}(\theta), 1 - \varepsilon_{\text{low}}, 1 + \varepsilon_{\text{high}}\right)\hat{A}_{i,t}\right) \tag{8}$$

As in DAPO, the loss of DCPO is not constrained by the KL divergence. For the lower and upper clipping bounds, we used the dynamic clipping bounds introduced in Section 2.

## 4 EXPERIMENTAL SETUP

**Models and Datasets** We experimented with four different models, Qwen2.5-Math-1.5B-Instruct, Qwen2.5-3B (common base), Qwen2.5-Math-7B (math base) and Qwen2.5-14B (common base) (Yang et al., 2024a;b), and used a combined training data set of approximately 25k mathematical problems, which combines the DAPO-Math-17K corpus (Yu et al., 2025) with levels 3-5 of the MATH subdataset (Hendrycks et al., 2021b; Liu et al., 2025). All problems used for training and evaluation were processed using the Qwen-Math template (see Section A.8).

**Evaluation** We evaluated our models on four widely used mathematical reasoning benchmarks: **MATH500** (Hendrycks et al., 2021a), **AMC23** (Ouyang et al., 2022), **AIME24** (li2), and AIME25 (MAA). For MATH500, we only report Avg@1 due to its larger size (500 problems). For the smaller AMC23 (40 problems), AIME24 (30 problems), and AIME25 (30 problems) datasets, we report both Avg@1 and Avg@32 to ensure a robust and comprehensive evaluation.

- **Avg@1**: This metric represents the standard accuracy achieved using greedy decoding. Measures the performance of the best prediction of the model.

- **Avg@32**: This metric calculates the average accuracy of 32 sampled responses per problem, using a temperature of 1.0 and top_p of 1.0. This metric provides insight into the robustness and stability of the trained policy distribution.

**Baseline Settings** We used GRPO and DAPO as baselines and added GSPO to Qwen2.5-Math-7B and Qwen2.5-14B as additional baselines. The other settings, such as the different clipping thresholds $\epsilon_*$, codebase, and other common training hyperparameters, are described in the Appendix A.9.

## 5 RESULTS AND ANALYSIS

### 5.1 MAIN RESULTS

Table 1: Performance across benchmarks. Avg@1: greedy decoding; Avg@32: sampling 32 times. Boldface shows the best values, and $\Delta$ shows the difference from the baseline. Red indicates DCPO outperforms the baseline.

| Model | MATH500 (Avg@1) | AMC23 (Avg@1/32) | AIME24 (Avg@1/32) | AIME25 (Avg@1/32) | **Average** (Avg@1/32) |
|---|---|---|---|---|---|
| Qwen2.5-Math-1.5B-Instruct | | | | | |
| base | 73.6 | 57.5/49.4 | 10.0/10.0 | 3.3/6.1 | 36.1/21.8 |
| GRPO | **77.2** | 70.0/68.4 | 16.7/14.0 | **20.0/13.5** | 46.0/32.0 |
| DAPO | 76.0 | **80.0**/70.6 | **20.0**/13.5 | 14.4/12.5 | 46.5/32.4 |
| DCPO | 77.2 | 75.0/**70.8** | **20.0**/15.6 | 16.7/12.1 | **47.2/32.8** |
| $\Delta_{GRPO}$ | +0.0 | +5.0/+2.4 | +3.3/+1.6 | -3.3/-1.4 | +1.2/+0.8 |
| $\Delta_{DAPO}$ | +1.2 | -5.0/+0.2 | +0.0/+2.1 | +2.3/-0.4 | +0.7/+0.4 |
| Qwen2.5-3B[Base] | | | | | |
| base | 46.4 | 27.5/7.8 | 3.3/0.1 | 3.3/0.7 | 20.1/2.9 |
| GRPO | 69.2 | 62.5/51.6 | **10.0**/7.5 | 6.7/4.5 | 36.3/21.0 |
| DAPO | **72.4** | 57.5/54.0 | **10.0**/8.3 | 6.7/3.9 | 36.6/23.1 |
| DCPO | 71.2 | **62.5/55.8** | 3.3/7.5 | **10.0/4.7** | **37.6**/22.7 |
| $\Delta_{GRPO}$ | +2.0 | +0.0/+4.2 | -6.7/+0.0 | +3.3/+0.2 | +1.3/+1.7 |
| $\Delta_{DAPO}$ | -1.2 | +5.0/+1.8 | -6.7/-0.8 | +3.3/+0.8 | +1.0/-0.4 |
| Qwen2.5-Math-7B[Base] | | | | | |
| base | 50.4 | 40.0/19.5 | 13.3/6.0 | 3.3/1.5 | 28.4/9.3 |
| GRPO | 81.6 | 77.5/75.9 | 36.7/32.1 | 16.7/16.7 | 53.1/41.6 |
| DAPO | 83.0 | 72.5/**80.7** | 36.7/31.6 | **23.3**/14.9 | 53.9/42.4 |
| GSPO | **84.0** | 80.0/78.8 | 40.0/34.9 | 16.7/16.2 | 55.2/43.3 |
| DCPO | 82.5 | **82.6**/79.8 | **46.7/38.8** | 16.7/**17.2** | **57.1/45.2** |
| $\Delta_{GRPO}$ | +0.9 | +5.1/+4.9 | +10.0/+6.7 | +0.0/+0.5 | +4.0/+3.6 |
| $\Delta_{DAPO}$ | -0.5 | +10.1/-0.9 | +10.0/+7.2 | -6.6/+2.3 | +3.3/+2.8 |
| $\Delta_{GSPO}$ | -1.5 | +2.6/+1.0 | +6.7/+3.9 | +0.0/+1.0 | +1.9/+1.9 |
| Qwen2.5-14B[Base] | | | | | |
| base | 60.8 | 47.5/16.4 | 3.3/1.3 | 3.3/1.1 | 28.7/6.3 |
| GRPO | 81.2 | 75.0/65.6 | 13.3/17.6 | 13.3/10.5 | 45.7/31.3 |
| DAPO | 83.4 | **87.5/85.1** | 16.7/16.4 | 20.0/15.3 | 51.9/38.9 |
| GSPO | 78.6 | 77.5/75.0 | **23.3**/16.0 | 16.7/9.9 | 49.0/33.5 |
| DCPO | **84.6** | 85.0/79.9 | 20.0/**18.2** | **23.3/19.0** | **53.2/39.0** |
| $\Delta_{GRPO}$ | +3.4 | +10.0/+14.3 | +6.7/+0.6 | +10.0/+8.5 | +6.5/+7.7 |
| $\Delta_{DAPO}$ | +1.2 | -2.5/-5.2 | +3.3/+1.8 | +3.3/+3.7 | +1.3/+0.1 |
| $\Delta_{GSPO}$ | +6.0 | +7.5/4.9 | -3.3/+2.2 | +6.6/+10.1 | +4.2/+5.5 |

Table 1 summarizes the performance of different methods, including GRPO, DAPO, GSPO, and our proposed DCPO, on four math reasoning benchmarks: MATH500, AMC23, AIME24, and AIME25. We report both greedy decoding results (Avg@1) and sampling-based decoding performance (Avg@32), providing a comprehensive evaluation of the robustness and generalization of each method. Across all models scales, DCPO consistently demonstrates superior or comparable performance relative to GRPO, DAPO and (where available) GSPO, indicating its effectiveness as a general-purpose preference optimization technique in reinforcement learning of LLMs.

**Overall Superiority of DCPO**: On the Qwen2.5-Math-1.5B-Instruct, DCPO achieves the highest average scores (47.2/32.8), exceeding GRPO (+1.2/+0.8) and DAPO (+0.7/+0.4). On Qwen2.5-3B, DCPO yields (37.6/22.7), exceeding GRPO (+1.3/+1.7) and performing competitively with DAPO, with a notable improvement in Avg@1 (+1.0) and only a marginal gap in Avg@32 (-0.4). For Qwen2.5-Math-7B, DCPO attains (57.1/45.2), surpassing GRPO (+4.0/+3.6) and DAPO (+3.3/+2.8) and GSPO (+1.9/+1.9). On the largest tested model, Qwen2.5-14B, DCPO delivers a strong result of (53.2/39.0), offering substantial improvements over GRPO (+6.5/+7.7) and outperforming DAPO

by (+1.3/+0.1) and GSPO by (+4.2/+5.5) in Avg@1 and Avg@32. These results highlight DCPO's robust improvements across different model capacities and its superior alignment to the reward signal under both greedy and stochastic decoding paradigms.

**Robustness on AIME under Sampling (Avg@32)**: A particularly notable trend emerges on the AIME24 and AIME25 datasets, high-difficulty competition math benchmarks designed to stress-test a model's reasoning ability. Under Avg@32, which evaluates the model's ability to generate correct responses under high-entropy sampling, DCPO exhibits significantly enhanced robustness on the 7B and 14B scales: On AIME24-Avg@32, DCPO-7B achieves 38.8, outperforming GRPO (32.1), DAPO (31.6) and GSPO (34.9). On AIME25-Avg@32, DCPO-14B reaches 19.0, exceeding GRPO (10.5), DAPO (15.3) and GSPO (9.9). These gains suggest that DCPO not only improves average case performance, but also enhances the diversity and reliability of correct generations, which is particularly critical under sampling-based decoding where beam search or temperature-driven variability is present. The method's ability to optimize both reward fidelity and exploration under high entropy decoding highlights its practical value in real-world settings where multiple plausible outputs are sampled.

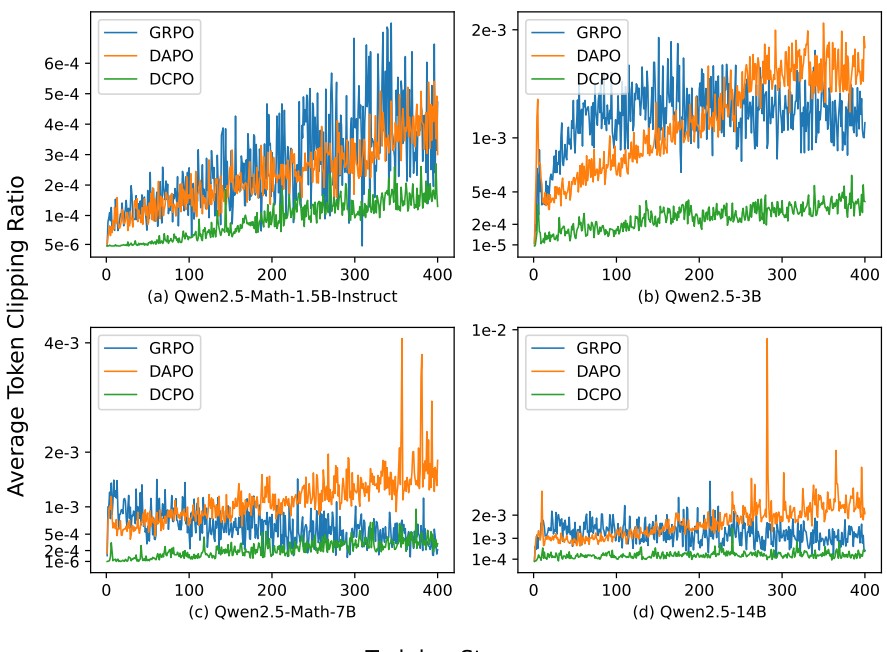

Figure 1: *TCR* across models and methods.

## 5.2 THE TOKEN CLIPPING RATIO (*TCR*)

We use the **Token Clipping Ratio** (*TCR*) as the proportion of tokens excluded from policy updates during back-propagation due to clipping. As shown in Figure 1, the behavior of the *TCR* varies significantly between different reinforcement learning methods.

$$\text{TCR} = Average \left( \sum_{m=1}^{N} \frac{\text{Number of clipped tokens in } micro_m}{\text{Total number of tokens in } micro_m} \right) \quad (9)$$

where $micro_m$ denotes the $m$-th microbatch and $N$ is the number of microbatches per training step.

In GRPO training, the *TCR* exhibits divergent trends based on the model scales. For smaller models (Qwen2.5-Math-1.5B and Qwen2.5-3B), the *TCR* increases with training steps, while for larger models (Qwen2.5-Math-7B and Qwen2.5-14B), it gradually decreases. In contrast, DAPO consistently shows an upward trajectory for all models. This indicates that as DAPO training progresses, an increasing number of tokens are clipped, leading to a higher proportion of policy

updates based on partial or truncated responses. However, both GRPO and DAPO exhibit significant fluctuations and occasional abnormal spikes in the *TCR*, which may introduce instability into training and potentially lead to entropy collapse. Meanwhile, the average *TCR* of GSPO based on Qwen2.5-Math-7B is greater than 11%, and based on Qwen2.5-14B is greater than 15%, which is much higher than the average *TCR* of token-level clipping methods and wastes more informative tokens.

In contrast, DCPO maintains a remarkably stable and efficient training state. Regardless of the size of the model or the training stage, the *TCR* for DCPO remains relatively constant and an order of magnitude lower than that of GRPO and DAPO. This lower and stabler *TCR* signifies that DCPO discards fewer tokens, allowing a greater portion of complete response sequences to contribute to the policy updates and freeing up more space for the model to explore the diversity of rare tokens. This finding highlights DCPO's superior sample efficiency compared to baselines while tightly controlling the space for outlier variation.

Taking Qwen2.5-Math-7B as an example, we observe that the proportion of token distribution quickly concentrates on high-confidence tokens: after approximately 60 steps, about 95% of generated tokens have a probability $(q(x) > 0.9)$; this fraction rises to about 97% after approximately 100 steps and continues to increase in later stages of training. As illustrated in Figure 4d, once the old probability satisfies $(q(x) \geq \frac{1}{1+\epsilon_{grpo}} \approx 0.83)$, the new probability $p(x)$ will participate in the model updates within the interval $[\frac{1}{1+\epsilon}, 1]$ without being discarded regardless of any clipping methods. Consequently, the vast majority of tokens lie in a region that is never clipped, which explains the low overall *TCR* observed in practice. Meanwhile, Figure 1 shows that the *TCR* grows during the later phases of most training tasks, indicating that an increasingly large proportion of low-probability tokens may be discarded and thus restricting the model's ability to explore the diversity of rare tokens.

Previous work (Wang et al., 2025) demonstrated that **high-entropy tokens** (precisely low-probability tokens) are the primary drivers of the model's emergence of reasoning capabilities. Since entropy is inversely proportional to probability, the rarer a token is, the more informational content it carries. Therefore, providing a broader admissible update range for such low-probability tokens enlarges the effective exploration space of the model, facilitating the acquisition of diverse reasoning capabilities.

These empirical and theoretical findings justify our proposed **dynamic-adaptive clipping** scheme, which relaxes clipping for tokens with $q(x) < \frac{1}{1+\epsilon}$ while preserving the safe interval for high-confidence tokens. By adaptively expanding the update interval for the high-entropy tail of the distribution, this method restores the model's capacity to learn from rare and informative tokens, thus enhancing its reasoning capabilities.

## 5.3 THE RESPONSE UTILIZATION RATIO (*RUR*)

During RLVR training, a policy update is contingent on nonzero gradients for responses. We quantify the efficiency of this process with a metric defined as the **Response Utilization Ratio** (*RUR*), which is the percentage of nonzero advantage of generated responses that participate in the policy update.

$$RUR = \frac{\text{Number of responses with non-zero advantage}}{\text{Total number of generated responses}} \times 100\% \qquad (10)$$

Table 2 summarizes the overall average *RUR* for responses generated during the training of GRPO is low, with an overall mean of only 43.8% in four different models, and that of GSPO is 45. 6% on the 7B and 14B scales. This indicates that more than half of the generated responses are discarded, highlighting the significant inefficiency in response utilization during the exploration of diversity. In contrast, our proposed DCPO method achieves an average *RUR* of approximately 70% after the first epoch. This represents a substantial improvement, with an absolute increase of 28% and a relative improvement of 64% over the GRPO's.

Figure 2 demonstrates that the *RUR* for GRPO training decreases significantly over time, regardless of the size or type of model. Notably, *RUR* drops sharply from above 90% to below 50%. For the Qwen2.5-Math-7B, for instance, the *RUR* plummets to as low as 30% after approximately 200 training steps. This finding suggests that GRPO's reliance on current advantage standardization results in a significant fraction of zero gradients, indicating that the model's diversity of exploration is notably inefficient in the later stages of training. In contrast, our proposed DCPO method, which employs the smooth advantage standardization, consistently maintains a high *RUR*. After the first epoch, the *RUR* quickly stabilizes around 70% and continues to improve with further training.

Table 2: Average *RUR* over 400 training steps.

| model | GRPO | GSPO | DCPO |
|---|---|---|---|
| Qwen2.5-Math-1.5B-Instruct | 45.6% | - | 67.1% |
| Qwen2.5-3B | 48.3% | - | 74.3% |
| Qwen2.5-Math-7B | 37.4% | 43.5% | 73.2% |
| Qwen2.5-14B | 43.9% | 47.6% | 72.4% |
| Average | 43.8% | 45.6% | 71.8% |

We observed that the *RUR* of DCPO exceeded 90% in the first few steps, but from the second epoch to the end of training, it remained around 70% and did not reach 100%. We analyze the reasons for these phenomena and demonstrate the potential value of the SAS mechanism.

First, the initial *RUR* exceeding 90% is largely due to the output of the early-stage model that does not comply with the required format, leading to negative rewards and inflated *RUR* values. This suggests that different models can quickly learn fixed answer patterns, so further emphasis should be placed on exploring the model's reasoning capabilities in subsequent training. Second, our analysis reveals two main factors that contribute to the consistently zero advantage, causing the *RUR* to plateau around 70% rather than reaching 100%: **(i)** Nearly half of these math problems are simple enough that the model consistently solves them correctly, and it is justifiable to exclude them from policy updates because the core capabilities of the model do not require further refinement in these instances. **(ii)** The remaining instances correspond to being consistently erroneous. This may arise from two factors: On the one hand, some problems are too challenging for the current model to solve, even with a correct ground-truth label. On the other hand, the DAPO-Math-17K dataset contains some items whose labels were converted to mislabeled integers, introducing potential "incorrect" labels. Discarding responses with zero advantage in these cases acts as a form of filtering, mitigating the risk of updating the model based on noisy or mislabeled data.

Unlike GRPO, DCPO and GSPO, DAPO discards responses with identical labels from generated data, making a direct *RUR* comparison difficult. Therefore, we evaluated the training efficiency of DAPO by comparing the required number of generated responses and the training time to complete the same number of update steps. To alleviate the low sampling efficiency caused by the low *RUR* of GRPO, we set the generation batch size at 1-4 times the training batch size and adjust it dynamically when training DAPO. Experiments show that, across four models, DAPO requires 3 to 5 times as many generated responses as DCPO to achieve the same number of parameter updates, resulting in at least doubling the training GPU hours without yielding better performance. Furthermore, we observed that DAPO's data utilization efficiency decreases with model size. For example, the proportion of responses used by DAPO based on Qwen2.5-14B dropped to less than 30% after about 300 steps. This indicates that the dynamic sampling approach of DAPO has low data utilization and training efficiency, and misses out on potential exploration of model diversity.

In summary, DCPO represents a significant advancement over both GRPO, DAPO and GSPO by substantially improving response utilization and training efficiency, fully utilizing sampled samples to provide more exploration space for model diversity, and improving performance.

### 5.4 ABLATION STUDIES

We conducted an ablation study on Qwen2.5-Math-7B to assess the contribution of each component in DCPO, using **Avg@32** as the evaluation metric. This metric highlights the robustness and stability of the learned policy distribution. To ensure fairness, each experiment modifies a single component of the baseline GRPO framework while keeping all other settings identical and removing the KL divergence term to align with DAPO, GSPO, and the full DCPO. The learning curves among 20 to 400 steps are shown in Figure 3.

The learning curves for the three individual components show that the model continuously improves its inference ability. The model surpasses GRPO and attains (or exceeds) the performance of DAPO and GSPO. Consequently, each modification contributes a distinct benefit, and their combination yields a substantial overall gain on baselines, demonstrating the effectiveness of the DCPO pipeline over baselines in reinforcement learning tasks. In the later stages of training, the performance of baselines plateaus and shows no further improvement, while DCPO continues to exhibit an upward

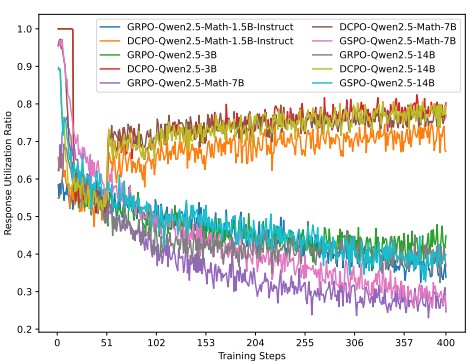

Figure 2: *RUR* progresses during training.

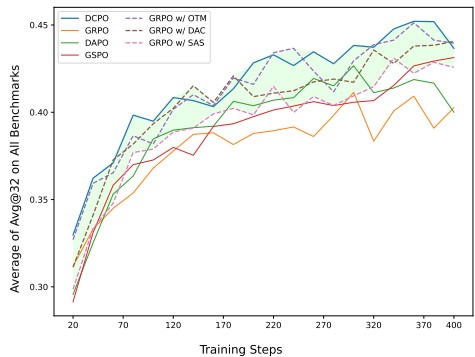

Figure 3: Ablation using the average Avg@32 based on Qwen2.5-Math-7B.

trend, both at the component level and overall. Ablation studies confirm that each component contributes positively to training efficiency, and their synergistic integration leads to cumulative gains and stability. Detailed performance is presented below.

(i) **Only-Token-Mean loss (OTM)**: When GRPO aggregated the loss within a single response rather than across the whole batch, the resulting model attains a performance that outperforms baselines, which suffer from lower data-utilization and fluctuating performance. (ii) **Smoothed Advantage Standardization (SAS)**: Replaces the traditional stepwise advantage with the cumulative smoothed advantage, which significantly improves the performance of GRPO and approaches the performance of DAPO. The gain demonstrates that aggregating reward statistics across training steps stabilizes optimization and enhances final performance. (iii) **Dynamic-Adaptive Clipping (DAC)**: Replacement of the fixed clipping boundary of GRPO with a probabilistically adaptive clipping boundary yields significant improvements. DAC itself surpasses GRPO, DAPO, and GSPO, and is more stable than using only OTM. (iv) **Combination of OTM + SAS + DAC (full DCPO)**: Combining these three mechanisms achieves the best and most stable overall results, significantly outperforming GRPO, DAPO, and GSPO. The synergy between OTM, SAS, and DAC demonstrates that each design choice contributes uniquely to data-efficient and stable reinforcement learning, improving model performance and robustness for large language models.

In general, the ablation study confirms that each component of DCPO contributes positively to overall performance, and their combination leads to substantial cumulative gains. The results validate the effectiveness of the proposed mechanisms in improving data efficiency and stability in reinforcement learning for LLMs.

## 6 CONCLUSION

In this work, we attempt to address key limitations of existing policy optimization methods: restricted token-level exploration due to fixed clipping in low-probability regions, and significantly low sample utilization due to the current step's reward standardization. To this end, we introduced **D**ynamic **C**lipping **P**olicy **O**ptimization (**DCPO**), a novel reinforcement learning pipeline that enhances the reasoning capabilities of large language models through two key innovations: (i) a **dynamically adaptive token-level clipping** mechanism that adjusts bounds based on old policy probabilities, enabling more efficient exploration of rare tokens with low probabilities, and (ii) a **Cumulative-Smooth Advantage Standardization** that aggregates the cumulative reward distribution across optimization steps, mitigating reward randomness from high-entropy sampling. Extensive experiments on four mathematical reasoning benchmarks with four models demonstrate DCPO's effectiveness. For example, the DCPO-7B variant achieved an Avg@32 score of 38.8 on the challenging AIME24 benchmark, outperforming GRPO (32.1) and DAPO (31.6). In addition, DCPO increased the response utilization ratio by 28% over GRPO, doubled the training efficiency compared to DAPO, and reduced the token clipping ratio by an order of magnitude. These results highlight DCPO's ability to significantly improve data utilization and diversity exploration in RLVR. Future work will explore the extension of DCPO to other domains, such as code generation and semantic reasoning.

# 7 REPRODUCIBILITY STATEMENT

We provide comprehensive details to ensure the reproducibility of our experiments. For the clipping thresholds of GRPO,GSPO and our DCPO and other hyperparameters, please see Section 4 and Appendix A.9 for detailed settings. We put our source code and data used in the experiments in an anonymous repository (https://anonymous.4open.science/r/DCPO-Anonymous-BE86-2509), and you can reproduce our work by following the instructions in the README file.

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

## A  APPENDIX

### A.1  LLMS USAGE STATEMENT

We just used LLMs (e.g., gpt-oss-120b (OpenAI, 2025b) and DeepSeek-R1) to review and correct grammar, capitalization, and sentence structure, primarily for Section 5 and the appendix. We also carefully reviewed the LLM's suggested revisions, rejected any inappropriate suggestions, and adjusted them to improve the content's readability.

### A.2  MAIN SYMBOL DEFINITIONS

| | |
|---|---|
| $G$ | Number of responses generated per prompt at each step. |
| $\theta$ | Parameters of the actor model. |
| $\pi_\theta(o_i\|q)$ | Probability of response $o_i$ given prompt $q$ under parameters $\theta$. |
| $\pi_{\theta_{old}}(o_i\|q)$ | Probability of response $o_i$ given prompt $q$ under previous parameters $\theta_{old}$. |
| $\pi_{ref}$ | Reference policy for KL divergence computation. |
| $\beta$ | Coefficient for KL divergence term (controls policy update vs. constraint trade-off). |
| $r_{i,t}(\theta)$ | Probability ratio $\frac{\pi_\theta(o_i^t\|q,o_i^{1:t-1})}{\pi_{\theta_{old}}(o_i^t\|q,o_i^{1:t-1})}$ for token $t$ in response $o_i$. |
| $\hat{A}_{i,t}$ | Standardized advantage estimate for token $t$ in response $o_i$. |
| $\|o_i\|$ | The length of the response in tokens. |
| $\mathbb{D}_{KL}(\pi_\theta\|\|\pi_{ref})$ | KL divergence between current and reference policies. |
| $\varepsilon$ | Probability ratio clipping threshold. |
| $\varepsilon_{low}$ and $\varepsilon_{high}$ | Lower/upper thresholds for probability ratio clipping. |
| $M$ | Maximum allowed response length. |
| $is\_equivalent(a, o_i)$ | Binary function indicating whether response $o_i$ matches answer $a$. |
| $\mu_{\text{new}}^i$ | Mean reward across $G$ responses generated at step $i$ (0 if $i = 0$). |
| $\sigma_{\text{new}}^i$ | Reward variance across $G$ responses at step $i$ (0 if $i = 0$). |
| $\mu_{\text{old}}^i$ | Mean reward across $G \cdot (i-1)$ responses from steps 1 to $i-1$ (0 if $i \le 1$). |
| $\sigma_{\text{old}}^i$ | Reward variance across $G \cdot (i-1)$ responses from steps 1 to $i-1$ (0 if $i \le 1$). |
| $\mu_{\text{total}}^i$ | Mean reward across $G \cdot i$ responses from steps 1 to $i$ (0 if $i = 0$). |
| $\sigma_{\text{total}}^i$ | Reward variance across $G \cdot i$ responses from steps 1 to $i$ (0 if $i = 0$). |

### A.3  PRELIMINARY

#### A.3.1  ADVANTAGE CALCULATION

In previous works, including GRPO and DAPO, the advantage $\hat{A}_{j,t}^i$ for the token $t$ in the response $j$ is calculated by standardizing the reward $R_j^i$ against the mean $\mu^i$ and standard deviation $\sigma^i$ of the rewards of the $G$ responses generated in the $i$-th step, as shown in Equation (11).

$$\hat{A}_{j,t}^i = \frac{(R_j^i - \mu^i)}{\sigma^i} \tag{11}$$

When the rewards for the same prompt are identical, responses with zero advantages do not contribute to model update, resulting in response waste.

### A.3.2 GRPO

Shao et al. (2024) proposed the **G**roup **R**elative **P**olicy **O**ptimization(**GRPO**), which first samples $G$ responses for each query, assigns rewards $R$ through a rule-based reward function and estimates token-level advantages as in Equation (11). Finally, the model parameters are updated using Equation (12), referred to as **S**equence **L**evel **M**ean loss (**SLM**) as:

$$\mathcal{T}_{\text{GRPO}}(\theta) = \frac{1}{G} \sum_{i=1}^{G} \frac{1}{|o_i|} \sum_{t=1}^{|o_i|} \min\left(r_{i,t}(\theta)\,\hat{A}_{i,t}, \text{clip}(r_{i,t}(\theta), 1-\varepsilon, 1+\varepsilon)\hat{A}_{i,t}\right) - \beta\mathbb{D}_{\text{KL}}(\pi_\theta\|\pi_{\text{ref}})$$

(12)

The first term averages token-level advantages over all tokens in a response and then over all $G$ responses in the batch. The $\mathbb{D}_{KL}$ penalty constrains the policy from deviating too far from a reference policy (e.g., a pre-trained model). The fixed clipping threshold $\epsilon$ prevents overly large updates by limiting the change in token probability ratios. This averaging strategy, though widely adopted, inherently dilutes per-response importance.

### A.3.3 DAPO

Similarly, Yu et al. (2025) propose **D**ynamic s**A**mpling **P**olicy **O**ptimization (**DAPO**), which observed that in GRPO, longer sequences exert a disproportionate influence on gradient update. They standardize token counts across the entire batch, leading to the **T**oken **L**evel **M**ean loss (**TLM**) as Equation (13).

$$\mathcal{T}_{\text{DAPO}}(\theta) = \frac{1}{\sum_{i=1}^{G}|o_i|} \sum_{i=1}^{G} \sum_{t=1}^{|o_i|} \min\left(r_{i,t}(\theta)\hat{A}_{i,t},\ \text{clip}\left(r_{i,t}(\theta), 1-\varepsilon_{\text{low}}, 1+\varepsilon_{\text{high}}\right)\hat{A}_{i,t}\right)$$

(13)

subject to $0 < |\{o_i \mid is\_equivalent(a, o_i)\}| < G$. It discards the prompts whose all responses share identical reward, and regenerates responses to maintain batch size, leading to severe data inefficiency.

### A.3.4 GSPO

Zheng et al. (2025) propose **G**roup **S**equence **P**olicy **O**ptimization (**GSPO**), arguing that token-level clipping can inject high-variance noise into gradient estimates. This noise tends to accumulate over long sequences and is further amplified by the clipping operation itself. GSPO alleviates the problem by replacing token-level clipping with sequence-level clipping, which yields a more stable and effective training signal.

$$\mathcal{T}_{\text{GSPO}}(\theta) = \frac{1}{G} \sum_{i=1}^{G} \min\left(s_i(\theta)\,\hat{A}_i, \text{clip}(s_i(\theta), 1-\varepsilon, 1+\varepsilon)\hat{A}_i\right)$$

$$\text{where } s_i(\theta) = exp\left(\frac{1}{|o_i|} \sum_{t=1}^{|o_i|} log\left(\frac{\pi_\theta(o_{j,t}|q)}{\pi_{\theta_{old}}(o_{j,t}|q)}\right)\right)$$

(14)

GSPO discards more than 10% of the responses according to sequence-level clipping, resulting in high data waste and training efficiency. Although GRPO discards responses with sequence-level high variance, it still retains a large number of responses with token-level high importance weight, which will be clipped by token-level clipping methods, and resulting in training instability.

### A.4 REWARD CALCULATION

To provide a verifiable and interpretable supervision signal, we adopt a rule-based reward model that evaluates each generated response based on the accuracy of the answers and the compliance of the format, as Equation (15). Specifically, a reward of +1 is assigned when both the format and the answer are correct; a reward of 0 is given when the format is correct but the answer is incorrect; and a reward of -1 is assigned when the format is incorrect.

$$R_j^i = \begin{cases} 1, & \text{correct format and correct answer} \\ 0, & \text{correct format but incorrect answer} \\ -1, & \text{incorrect format} \end{cases} \tag{15}$$

where $j$ is the index of the response group for a given problem in the $i$-th training step, and the reward is calculated based on the format and correctness of the response.

### A.5 Hypothetical Reasoning Process for Dynamic-Adaptive Clipping Bounds

Clipping the probability ratio is a crucial technique in reinforcement learning to stabilize policy optimization and prevent large, destabilizing parameter update. This mechanism is particularly important when using importance sampling to estimate expected values under a new policy from samples generated by an old policy.

The expected value of a function $f(x)$ under the new probability $p(x)$ can be rewritten as an expectation under the old probability $q(x)$ by importance sampling weight, as shown in equation 16:

$$\int f(x)p(x)\,\mathrm{d}x = \int f(x)\frac{p(x)}{q(x)}q(x)\,\mathrm{d}x = \mathbb{E}_{x\sim q}\left[f(x)\frac{p(x)}{q(x)}\right] \tag{16}$$

Although this estimator is unbiased, its variance can be significantly inflated, which is a common challenge in importance sampling. The true variance of the function $f(x)$ under the new policy is given by Equation (17), whereas the variance of the importance sampling estimator under the old policy is given by Equation (18):

$$\mathrm{Var}_{x\sim p}[f(x)] = \mathbb{E}_{x\sim p}\left[f(x)^2\right] - \left(\mathbb{E}_{x\sim p}[f(x)]\right)^2 \tag{17}$$

$$\begin{aligned} \mathrm{Var}_{x\sim q}\left[f(x)\frac{p(x)}{q(x)}\right] &= \mathbb{E}_{x\sim q}\left[\left(f(x)\frac{p(x)}{q(x)}\right)^2\right] - \left(\mathbb{E}_{x\sim q}\left[f(x)\frac{p(x)}{q(x)}\right]\right)^2 \\ &= \mathbb{E}_{x\sim p}\left[f(x)^2\frac{p(x)}{q(x)}\right] - \left(\mathbb{E}_{x\sim p}[f(x)]\right)^2 \end{aligned} \tag{18}$$

The difference between the importance sampling variance and the true variance highlights the potential for inflation, as shown in Equation (19):

$$\mathrm{Var}_{x\sim q}\left[f(x)\frac{p(x)}{q(x)}\right] - \mathrm{Var}_{x\sim p}\left[f(x)\right] = \mathbb{E}_{x\sim p}\left[f(x)^2(\frac{p(x)}{q(x)} - 1)\right] = \int f(x)^2(\frac{p(x)}{q(x)} - 1)p(x)\mathrm{d}x \tag{19}$$

To mitigate this difference, many prior works, including PPO (Schulman et al., 2017) and RLVR methods (DeepSeek-AI, 2024; Liu et al., 2025), employ a fixed symmetric clipping bound on the probability ratio $r(x) = \frac{p(x)}{q(x)}$, as defined in Equation (20):

$$|r(x) - 1| \leq \epsilon \tag{20}$$

However, a symmetric-fixed clipping bound inherently constrains the policy's capacity for exploration, particularly in regions where the old policy assigns a low probability mass. In such cases, the absolute bound for low probabilities is severely limited, restricting the potential for meaningful policy updates. While methods like DAPO attempt to address this by introducing asymmetric clipping bounds, the fundamental limitation of a non-adaptive constraint remains.

In addition, recent studies such as Yue et al. (2025) suggest that the reasoning paths leveraged by RLVR models are largely present within the sampling distribution of the base model. This is further evidenced by the minimal parameter shift observed during off-policy batch update, as reflected in the small fraction of clipped ratios. Given this stability, we propose a more practical alternative

to constrain the probability ratio $r(x)$ through the dynamic-adaptive mechanism by including the probability in the restriction as in Equation (21), replacing the conventional fixed clipping approach of Equation (20).

$$|(\frac{p(x)}{q(x)} - 1)p(x)| \leq \epsilon \tag{21}$$

Expanding the substitution of $p(x) = r(x)q(x)$ into the inequality of a previous equation yields the expression in Equation (22).

$$-\epsilon \leq (r(x) - 1)r(x) * q(x) \leq \epsilon \tag{22}$$

For this clipping ratio to be mathematically valid and meaningful, we must adhere to the fundamental constraints of probability distributions and their ratios. Specifically, as shown in Equation (23), the probabilities $p(x)$ and $q(x)$ must be non-negative and less than or equal to 1. Consequently, the probability ratio $r(x)$ must also be non-negative, and the clipping hyper-parameter $\epsilon$ must be non-negative.

$$\begin{cases} 0 \leq p(x) \leq 1 \\ 0 \leq q(x) \leq 1 \\ 0 \leq r(x) \\ 0 \leq \epsilon \end{cases} \tag{23}$$

To ensure the validity and practical applicability of the clipping bound derived from Equation (22), we define the necessary conditions for its bounds as outlined in Equation (24).

$$\begin{cases} low_{ge}(q(x), \epsilon) = 0.5 - \frac{1}{2}\sqrt{1 + \frac{4\epsilon}{q(x)}} \\ high_{le}(q(x), \epsilon) = 0.5 + \frac{1}{2}\sqrt{1 + \frac{4\epsilon}{q(x)}} \\ low_{le}(q(x), \epsilon) = 0.5 - \frac{1}{2}\sqrt{1 - \frac{4\epsilon}{q(x)}}, & \text{S.t.} \quad q(x)^2 - 4q(x) * \epsilon \geq 0 \\ high_{ge}(q(x), \epsilon) = 0.5 + \frac{1}{2}\sqrt{1 - \frac{4\epsilon}{q(x)}}, & \text{S.t.} \quad q(x)^2 - 4q(x) * \epsilon \geq 0 \end{cases} \tag{24}$$

As we can get from Equation (22) to Equation (24), the probability ratio $r(x)$ should be met by the following conditions:

$$\begin{cases} low_{ge}(q(x), \epsilon) \leq 0 \leq r(x) \leq high_{le}(q(x), \epsilon), & \text{when } \text{Var}_{x \sim q}\left[f(x)\frac{p(x)}{q(x)}\right] \geq \text{Var}_{x \sim p}[f(x)] \\ r(x) \leq low_{le}(q(x), \epsilon) \cup high_{ge}(q(x), \epsilon) \leq r(x), & otherwise \end{cases} \tag{25}$$

For the clipping bounds, we consider the overlapping region centered around the ratio value of 1 to minimize the discrepancy of bounds between the two conditions and to maintain consistency with the conventional fixed bound of GRPO, and define the following conditions to ensure the validity of the clipping bounds:

$$high_{ge}(q(x), \epsilon) \leq r(x) \leq high_{le}(q(x), \epsilon)$$
$$\text{S.t.} \quad q(x)^2 - 4q(x) * \epsilon \geq 0 \tag{26}$$

For $low_{le}$ and $high_{ge}$, to ensure that the square number is not less than zero, we use the method of comparing with 0 and taking the maximum value of the approximation like $\max(1 - \frac{4\epsilon}{q(x)}, 0)$. Finally, we can get the following conditions, as **dynamic-adaptive clipping bounds for importance sampling**:

$$0.5 + \frac{1}{2}\sqrt{\max\left(1 - \frac{4\epsilon_{low}}{q(x)}, 0\right)} \leq r(x) \leq 0.5 + \frac{1}{2}\sqrt{1 + \frac{4\epsilon_{high}}{q(x)}} \tag{27}$$

For the upper and lower clipping bounds in Inequality 27, we set different upper and lower clipping thresholds $\epsilon_{high}$ and $\epsilon_{low}$ hyperparameters to align the clipping bounds with the GRPO at specific points.

### A.6 The Differences Between Dynamic-Adaptive and Fixed Clipping

To ensure that our dynamic-adaptive clipping bounds align with the fixed symmetric clipping bounds at the boundary points $(\frac{1}{1+\epsilon}, 1)$ and $(1, 1 - \epsilon)$, as shown in Figure 4d, we derive the specific values for $\epsilon_{\text{low}}$ and $\epsilon_{\text{high}}$ by solving the equations in Equation (28).

$$
\begin{cases}
q(x) = \frac{1}{1+\epsilon} \\
1 = 0.5 + \frac{1}{2}\sqrt{1 + \frac{4\epsilon_{high}}{q(x)}}
\end{cases}
\qquad
\begin{cases}
q(x) = 1 \\
1 - \epsilon = 0.5 + \frac{1}{2}\sqrt{max(1 - \frac{4\epsilon_{low}}{q(x)}, 0)}
\end{cases}
\tag{28}
$$

Typically, in previous practices(e.g., GRPO), the fixed clipping threshold $\epsilon$ is set to 0.2, and we can get our clipping hyperparameters to be approximately $\epsilon_{low} = 0.16$ and $\epsilon_{high} = 0.2$, while $\epsilon_{low} \neq \epsilon_{high}$ has a different purpose from the asymmetric fixed clipping setting in DAPO.

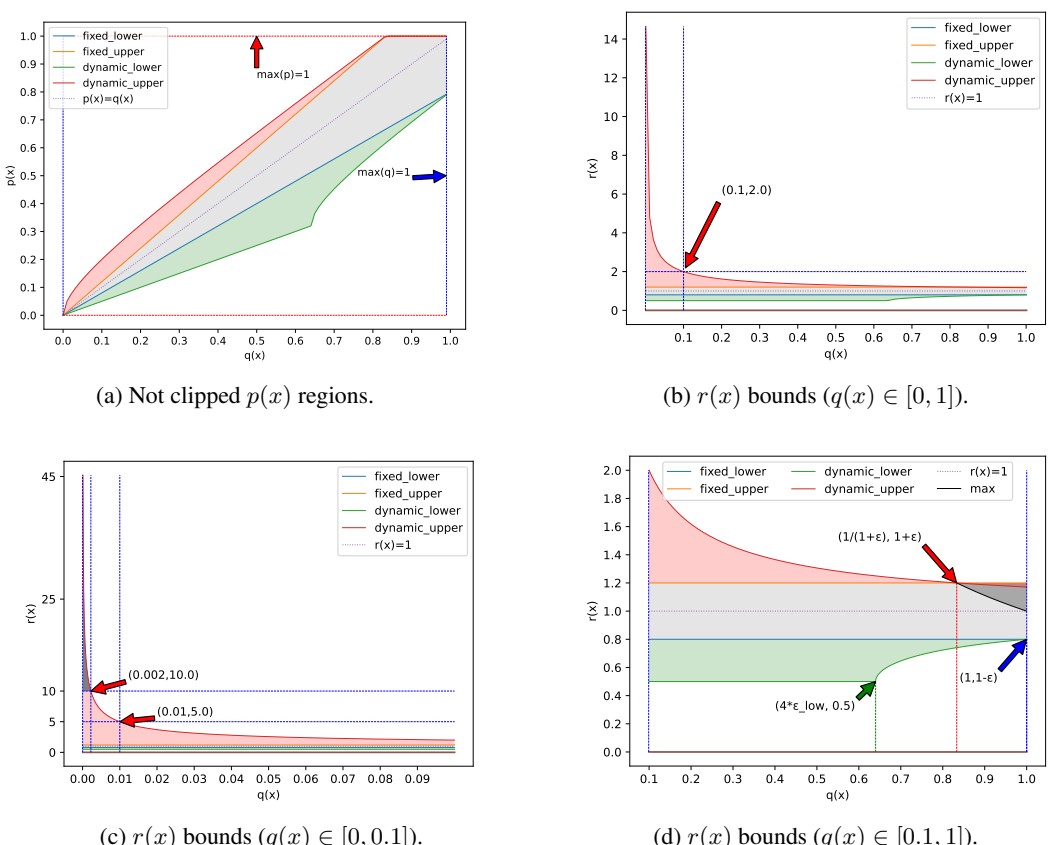

(a) Not clipped $p(x)$ regions.

(b) $r(x)$ bounds ($q(x) \in [0, 1]$).

(c) $r(x)$ bounds ($q(x) \in [0, 0.1]$).

(d) $r(x)$ bounds ($q(x) \in [0.1, 1]$).

Figure 4: Clipping bound comparisons. Lines show bounds for fixed clipping ($\varepsilon = 0.2$) vs. dynamic-adaptive clipping ($\varepsilon_{\text{low}} = 0.16, \varepsilon_{\text{high}} = 0.2$).

**Dynamic versus fixed clipping**: Figure 4 illustrates how the bound behaves under the fixed clipping scheme used in GRPO ($[1 - \varepsilon, 1 + \varepsilon]$) and under our probability-aware adaptive clipping. For a token with an old probability $q(x)$, the fixed bound provides a constant relative clipping width $2\varepsilon$ for $r(x)$. Consequently, as $q(x)$ decreases, the absolute exploration space for the new probability

$q(x)$ allowed shrinks linearly (Figure 4a). In contrast, the relative width of dynamic adaptation is inversely proportional to $q(x)$, so the exploration space for $q(x)$ remains roughly constant throughout the probability range.

**Effect of decreasing** $q(x)$: When $q(x)$ becomes small, the admissible interval $[1 - \varepsilon, 1 + \varepsilon]$ of the fixed clipping scheme is independent of $q(x)$, so the model's ability to explore low-probability tokens does not improve, while the dynamic-adaptive bound expands rapidly (Figure 4b), resulting in a larger relative space for rare tokens, which is exactly the behavior required for efficient RL.

**Behavior for extremely low probabilities**: Figure 4c shows the clipping interval when $q(x) \in [0, 0.1]$. As $q(x) \to 0$, the upper bound for $r(x)$ of dynamic-adaptive clipping increases rapidly inversely proportional to the square root, while the corresponding value for fixed clipping remains constant. Inspired by the dual-clipping strategy(Ye et al., 2020), we enforce a hard ceiling $r_{\max} = 10$ to prevent the resulting instabilities (gradient vanishing, explosion, etc.). The gray region in Figure 4c corresponds to approximately $q(x) \leq 2 \times 10^{-3}$, where the upper limit of adaptive clipping is set to $r_{\max}$.

**Upper and lower bounds**: For the lower bound, when $q(x) \leq 4\varepsilon$ the dynamic-adaptive clipping yields $dynamic_{low} = 0.5$, whereas the fixed clipping retains the constant $fixed_{lower} = 1 - \varepsilon = 0.8$. When $q(x)$ increases beyond this threshold, the dynamic-adaptive lower bound smoothly contracts towards the fixed value. Similarly, the dynamic-adaptive upper bound exceeds the fixed bound whenever $q(x) < \frac{1}{1+\varepsilon_{\text{high}}}$, and merges with it for larger $q(x)$. Thus, the adaptive interval $[dynamic_{lower}\left(q\left(x\right), \epsilon_{low}\right), dynamic_{upper}\left(q\left(x\right), \epsilon_{high}\right)]$ provides a probability-dependent widening of the admissible region for rare tokens while preserving the tightness of the fixed interval for common tokens (Figure 4d).

**Positive versus negative advantage**: The loss in Equation (8) contains both a positive and a negative advantage term. During the operation with minimum loss value, the ratio $r(x)$ for a positive advantage is clipped in the interval $[0, dynamic_{upper}(q(x), \epsilon_{high})]$. For a negative advantage, the clipping interval $[dynamic_{lower}(q(x), \epsilon_{low}), \infty)$. Due to an excessively large $r(x)$, which can destabilize training, we limit the ratio for both positive and negative advantages at $r_{\max} = 10$. This precaution is inspired by Ye et al. (2020) and ensures that the adaptive limits do not produce excessively high importance weights.

In short, regardless of positive or negative advantages, compared to fixed clipping, dynamic-adaptive clipping provides the model with more space to explore diversity when rare tokens have a low probability.

### A.7    THE DEFINITIONS OF THE ADVANTAGE STANDARDIZATION

Let $i$ denote the $i$-th iteration where prompts are used to generate responses and update the model parameters. We define the following statistics for reward normalization.

For the average for the rewards of responses, we define the equations as the group of functions as in Equation (29):

$$
\begin{cases}
\mu_{new}^{i} = \frac{1}{G} \sum_{j=1}^{G} R_j^i \\
\mu_{old}^{i} = \frac{1}{G*(i-1)} \sum_{k=1}^{i-1} \sum_{j=1}^{G} R_j^k \\
\mu_{total} = \frac{1}{G*i} \sum_{k=1}^{i} \sum_{j=1}^{G} R_j^k \\
\qquad = \frac{1}{i} \left( \mu_{new}^i + (i-1) \mu_{old}^i \right)
\end{cases}
\tag{29}
$$

For the variance for the rewards of the responses, we define the equations as the group of functions as in Equation (30):

$$
\begin{cases}
\sigma_{new}^i = \sqrt{\dfrac{1}{G}\sum_{j=1}^{G}\left(R_j^i - \mu_{new}^i\right)^2} \\[2ex]
\sigma_{old}^i = \sqrt{\dfrac{1}{G*(i-1)}\sum_{k=1}^{i}\sum_{j=1}^{G}\left(R_j^k - \mu_{old}^k\right)^2} \\[2ex]
\sigma_{total}^i = \sqrt{\dfrac{1}{G*i}\sum_{k=1}^{i}\sum_{j=1}^{G}\left(R_j^k - \mu_{total}^k\right)^2} \\[2ex]
\qquad\;\; = \sqrt{\dfrac{1}{i}\left({\sigma_{new}^i}^2 + (i-1){\sigma_{old}^i}^2 + \dfrac{i-1}{i}\left(\mu_{old}^i - \mu_{new}^i\right)^2\right)}
\end{cases}
\tag{30}
$$

To reduce resource usage, for the calculation of the value of $\sigma_{total}^i$ and $\mu_{total}^i$ in Equation (30) and Equation (29), we use equivalent calculations of $\mu_{new}^i$, $\sigma_{new}^i$, $\mu_{old}^i$ and $\sigma_{old}^i$ instead of direct calculations of original rewards.

## A.8 THE TEMPLATE FOR EXPERIMENTS

We use the following template for all experiments in this paper, which is based on the Qwen-Math template (Yang et al., 2024b).

> **Qwen-Math Template**
>
> ```
> <|im_start|>system
> Please reason step by step, and put your final answer within  \boxed{}. <|im_end|>
> <|im_start|>user
> {problem}
> <|im_end|>
> <|im_start|>assistant
> ```

## A.9 TRAINING SETTINGS

We set a symmetric clipping threshold of $\epsilon = 0.2$ in GRPO and low/high clipping thresholds $(0.2, 0.28)$ with an additional 512 token soft-punish cache and the maximum probability ratio 10 for negative advantage for DAPO, low/high clipping thresholds $(3e-4, 4e-4)$ for GSPO. For our proposed DCPO, we set low/high clipping thresholds as $(0.16, 0.2)$, preserving the same clipping points $(q(x), r(x)) = (\frac{1}{1+\epsilon}, 1+\epsilon)$ and $(q(x), r(x)) = (1, 1-\epsilon)$ as GRPO (Figure 4d), and set the maximum probability ratio at 10 for both positive and negative advantages, different from DAPO. The threshold values in the baselines are used in many previous works. The KL penalty coefficient $\beta$ is set to 0.001 for GRPO.

Training was conducted for 400 steps with a batch size of 512 for response generation and a mini-batch size of 32 for parameter update. We used $temperature = 1$ and $top\_p = 1$ to generate $G{=}16$ responses for each problem, while the maximum length of the prompt was set to 1,024 and that of response was set to 3,072, which is much smaller than the maximum length of 20k used in the original DAPO.

We adapted our training codebase from Verl (Sheng et al., 2024) on 32xH20 GPUs. Unless stated otherwise, the hyperparameters matched those of DAPO. For loss computation, we retained the original $\frac{1}{G}$ or $\sum_{i=1}^{G}\frac{1}{|o_i|}$ weighted average formulation for baselines, instead of the approximate weighted calculation used in Verl, to ensure compatibility with different microbatch sizes.

## A.10 DETAILED PERFORMANCE OF MODELS

**Avg@1 Trend** Figure 5 shows the performance of the model using the **average** of Avg@1 metric. This metric is calculated by averaging the greedy decoding (Avg@1) results on the MATH500,

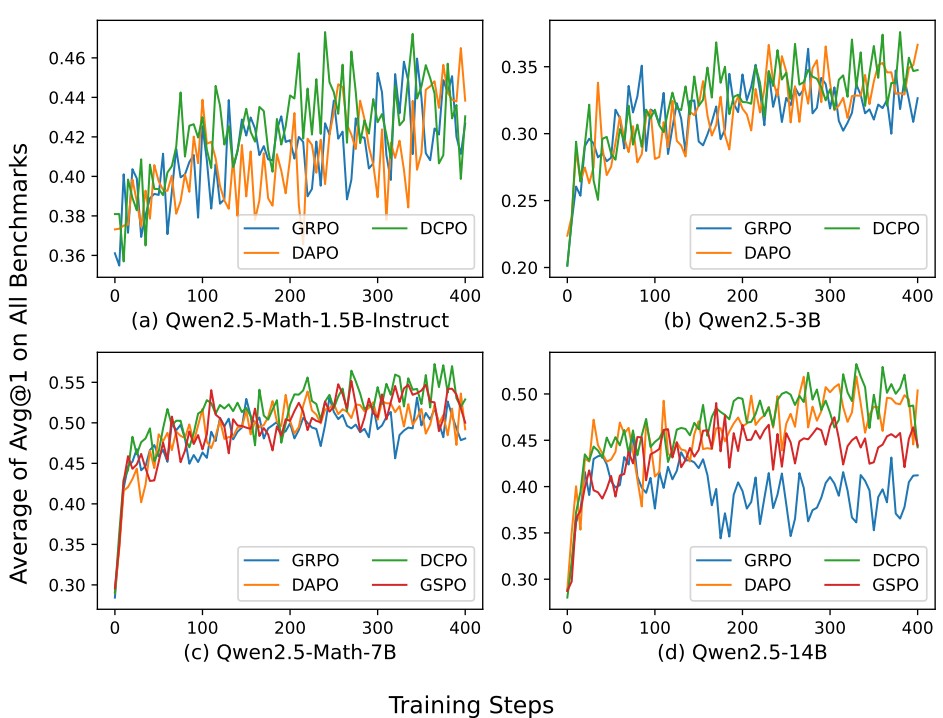

Figure 5: Avg@1 performance across benchmarks

AMC23, AIME24, and AIME25 benchmarks and represents the best performance of the model. As can be seen in the figure, DCPO consistently outperformed GRPO and DAPO across all four models, and outperformed GSPO on the two models where GSPO was evaluated. Furthermore, its overall performance was more stable than the baseline, without the significant performance drop seen in GRPO or the significant fluctuations seen in DAPO. This suggests that DCPO can stimulate stronger model reasoning capabilities during training, providing greater potential for the model to explore even better performance.

**Avg@32 Trend**    Figure 6 shows the **average** of Avg@32 performance of DCPO with GRPO, DAPO and (where available) GSPO at different model sizes on AMC23, AIME24, and AIME25 benchmarks, clearly demonstrating DCPO's exceptional robustness and stability. In contrast, GRPO exhibits significant fluctuation and occasional performance collapse. By GRPO, the Qwen2.5-14B model even experiences a significant performance drop after approximately 170 training steps. Across all evaluated models (from 1.5B to 14B), the DCPO performance trajectory exhibits a consistently upward trend with minimal fluctuation. Even in the early stages of training, DCPO maintains the fastest growth rate and maintains its upward trend. Furthermore, compared to DAPO, DCPO achieves superior performance while saving half of training time and resources. These experimental results demonstrate the effectiveness and robustness of dynamic-adaptive clipping and smoothing advantage standardization to improve the exploration at the response level and the label level of diversity.

**Entropy Trend**    Figure 7 shows that GRPO exhibits a sharp entropy decay after an initial warm-up phase and quickly stabilizes in a flat, low-entropy range. This entropy collapse reflects a rapid loss of policy randomness, reducing the diversity of sampled responses, and thus limiting subsequent policy optimization.

The behavior of entropy for different optimization methods is model-dependent. In the Instruct model (Qwen2.5-Math-1.5B), DAPO, DCPO, and GRPO follow similar declining trajectories. In contrast, for the larger base models (Qwen2.5-3B, Qwen2.5-Math-7B, and Qwen2.5-14B), the trends differ significantly. DAPO maintains the highest entropy value, showing a clear upward trend accompanied by large fluctuations. On Qwen2.5-Math-7B, the entropy value of GSPO is similar to that of DCPO, and the average metric performance is also comparable. Meanwhile, on Qwen2.5-14B,

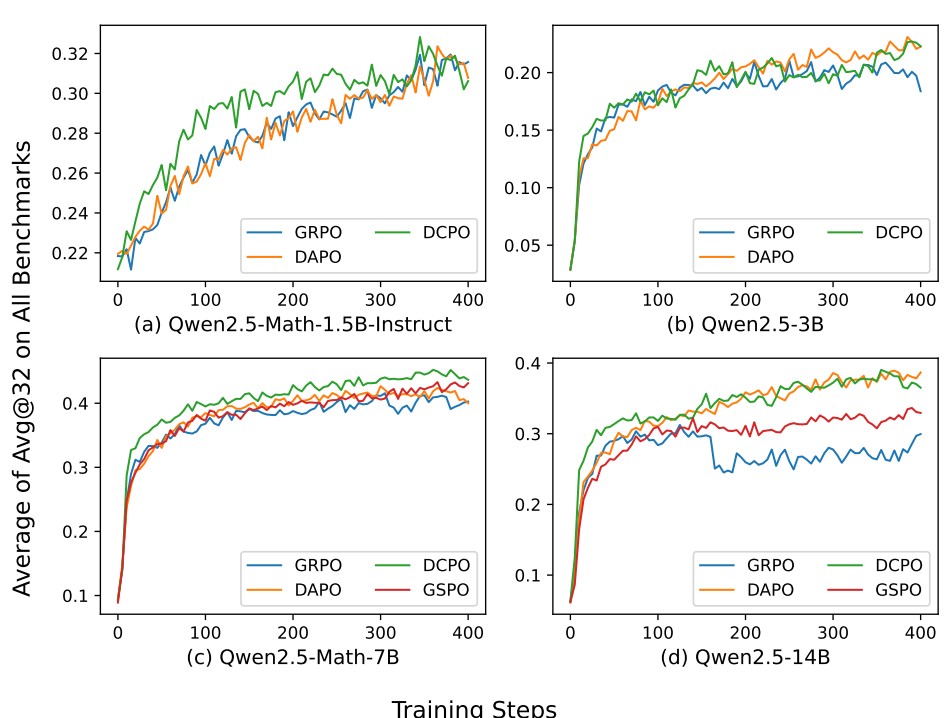

Figure 6: Avg@32 performance across benchmarks.

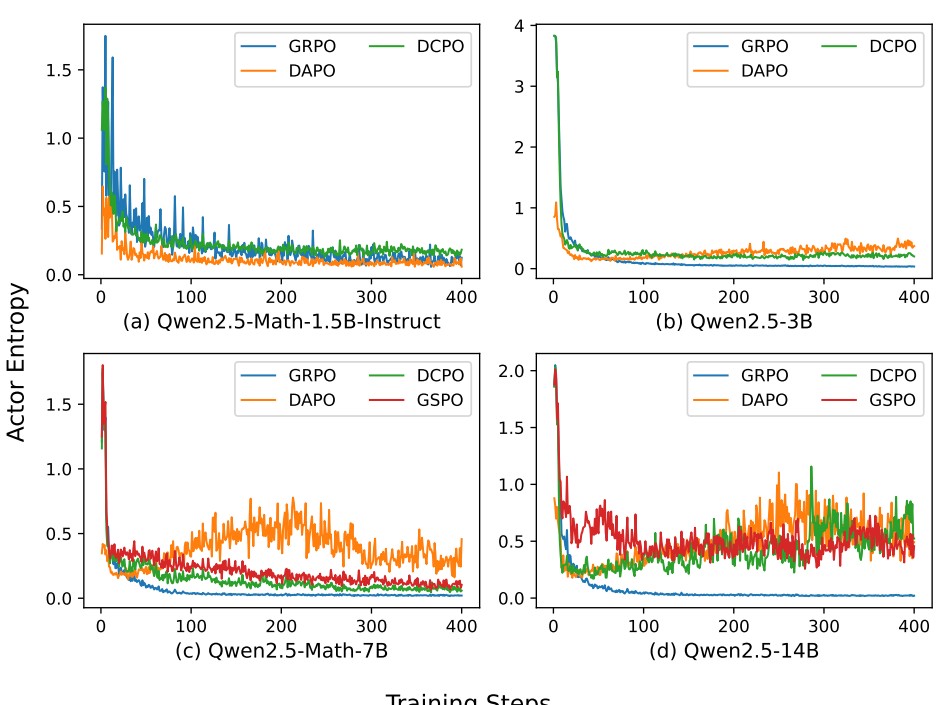

Figure 7: The entropy of the models during Training.

the performance of GSPO shows a fluctuating downward trend, resulting in poor performance. In contrast, DCPO reaches a moderate entropy level: its curve is significantly smoother and remains confined to a bounded band between the low-entropy region of GRPO and the high-variance region of DAPO.

These observations suggest a U-shaped relationship between entropy and training performance. Excessively low entropy deprives the learner of exploratory signal, leading to premature convergence, while excessively high or unstable entropy compromises training stability. DCPO maintains entropy within a moderate and well-balanced range, promoting convergence and sufficient exploration, a benefit that is particularly pronounced as model capacity grows. Therefore, DCPO's entropy regulation mechanism may underlie its superior inference performance and enable more robust policy learning across models of varying sizes.

