# OpenReview forum: "DCPO: Dynamic Clipping Policy Optimization"
_ICLR.cc/2026/Conference — Submitted to ICLR 2026_

### Official Review · Reviewer_s7dw · 2025-10-30

**Soundness:** 2
**Presentation:** 3
**Contribution:** 2
**Rating:** 4
**Confidence:** 3

**Summary:**

This paper introduces Dynamic Clipping Policy Optimization (DCPO), a novel reinforcement learning method for large language models that addresses the zero-gradient problem in existing approaches like GRPO. By implementing adaptive token-level clipping bounds and smooth advantage standardization, DCPO enhances exploration and response utilization, achieving good performance across multiple benchmarks—including some improvements over GRPO, DAPO, and GSPO—while boosting training efficiency and reducing token clipping rates.

**Strengths:**

- This paper is easy to read and easy to follow
- This paper focuses on an important topic in large language model reasoning, aiming at enhancing the reasoning capabilities of existing LLMs via some specially designed approaches
- The idea of dynamic clipping seems to be novel, and the analysis process is interesting

**Weaknesses:**

- (major) The main experiments are conducted only on Qwen models, raising doubts about its general benefits when using other LLMs as base models. Qwen series are known to suffer from dataset leakage, making it unclear whether the reported performance in tasks like AIME is reliable or trustworthy
- (major) No code, no homepage, and no models are available, making it unclear whether the results can be reproduced. The reviewer deems it important to open-source the code, datasets, and models in LLM research
- (major) The ablation study part is limited. The authors only conduct experiments on Qwen2.5-Math-7B and report Avg@32 on all benchmarks, and the detailed results on each benchmark should appear in the appendix. Furthermore, based on Figure 3, GRPO w/OTM loss achieves quite similar performance as DCPO, indicating that other components used in DCPO are less effective
- (major) DCPO contains numerous components, but many of them are not novel, either simply adapt from some prior works or just make some minor modifications. I hence cannot say that the technical contributions of DCPO are convincing enough to be accepted in this venue. For example, the authors adopt the dual clipping method from (Ye et al, 2020), the OTM loss is a minor modification compared to SLM or TLM loss
- (major) The performance improvement of DCPO is only marginal on numerous tested tasks (e.g., the performance of DCPO is quite close to that of DAPO). The authors claimed that they achieved state-of-the-art results. The reviewer cannot agree with that.
- (minor) This paper can benefit greatly from including more LLM-related references, especially those that investigate LLM math reasoning or improving DAPO/GRPO. This is a fast-growing research area, and there are numerous papers that do similar things. The authors should include more discussion in the manuscript
- (minor) Figure 1 is hard to interpret, the y-axis has quite confusing scales

**Questions:**

The performance of DAPO seems to be inferior to its performance in the original paper, have you checked that? Can the performance of DAPO become stronger with more training steps?

---

> ### Author Response · Authors · 2025-11-22
> **The 1/2 part of the reply**
>
> We thank the s7dw for the detailed and constructive feedback. Below, we address each concern point-by-point.
>
> ## W-1: Generalization beyond Qwen & dataset leakage concerns
> We acknowledge the concern regarding Qwen-series contamination, as highlighted by  [Reasoning or Memorization...](https://arxiv.org/abs/2507.10532) . Importantly:
>
> - In this study, **AIME24 and AIME25 showed no signs of contamination
>   1. **AIME25** was released later than the Qwen2.5 model, so contamination is unlikely.
>   2. we **will not** over-interpret the AMC/MATH results; these results are only for supplementary reference.
> - Following the suggestions in the contamination paper, we added **three supplementary benchmarks with no known risk of leakage**: **LiveMath, MinervaMath, and Olympiad**.
>
> **Table 1: Supplementary Avg@32 Performance Comparison (Qwen2.5-Math-7B)**
> |Method|Avg|AIME24|AIME25|AMC23|LiveMath|MinervaMath|Olympiad|
> |--|--|--|--|--|--|--|--|
> |DCPO|37.2|38.8|17.2|79.8|11.3|33.9|42.2|
> |GSPO|36.1(-1.2)|34.9(-3.9)|16.1(-1.1)|78.8(-1.0)|10.3(-1.0)|32.8(-1.1)|43.4(+1.2)|
> |DAPO|35.8(-1.4)|34.9(-3.9)|15.5(-1.7)|77.6(-2.2)|10.9(-0.4)|33.8(-0.1)|42.3(+0.1)|
> |GRPO|34.7(-2.6)|32.1(-6.7)|16.7(-0.5)|75.9(-3.9)|12.0(+0.7)|31.4(-2.5)|39.8(-2.4)|
>
> We also experimented with LLaMA-based methods. However, these experiments experienced performance degradation during training. Due to limited resources, we were unable to conduct further validation.
>
> ## W-2: Reproducibility—code, configuration, and models
> Our paper includes a complete reproducibility statement (**Section 7**) and an anonymous git containing: All training scripts,Data and code. The exact hyperparameters used in all experiments.
>
> All experiments can be started step-by-step by following the steps in the "[README.md](https://anonymous.4open.science/r/DCPO-Anonymous-BE86-2509/README.md)". **The main hyperparameters are listed in Section 4 and Appendix A.9.**
>
> Upon acceptance, we will publicly release the full codebase and trained models.
>
> ## W-3: Ablation study concerns & the role of OTM
>
> Due to computational resource constraints, we used Qwen2.5-Math-7B as the base model for our ablation experiments. Although GRPO and OTM appear to perform similarly in some points, **each ablation experiment result confirms**:
>
> - **DAC** improves token-level update utilization by reducing the token pruning rate (TCR) by an order of magnitude.
> - **SAS** addresses the instability caused by randomness by sampling the policy, improving response-level utilization (RUR) by **64%**.
> - **OTM** normalizes the advantage without distorting the DAC + SAS distribution.
>
> Together, they constitute a **stable and synergistic algorithm**. GRPO performance is not contributed by the individual components; its performance stems from the interaction between DAC, SAS, and OTM.

---

> ### Author Response · Authors · 2025-11-22
> **the 2/2 part of the reply**
>
> ## W-4: Novelty of DCPO relative to prior clipping methods
>
> The main contributions of DCPO **(including DAC and SAS) are all proposed for the first time**. OTM is also an optimization and improvement on the shortcomings of SLM and TLM. We believe that these are innovative contributions, and experimental results prove the effectiveness of their innovation.
>
> We clarify here that DCPO **does not** reuse dual clipping (Ye et al., 2020):
>
> - Based on a fixed symmetric threshold, dual clipping only restricts the maximum clipping bound of negative-advantage samples  to prevent over-updating of negative-advantage samples within the fixed clipping bounds.
> - **DCPO's DAC** is derived from the importance sampling principle, ultimately resulting in a method that **adaptively adjusts the boundary based on token's probabilities**, rather than a globally fixed clipping bound.
> - Inspired by dual clipping, we set a **max value boundary for all ( both positive and negative advantage ) samples to prevent over-updating within dynamic clipping boundaries**.
>
> Similarly, OTM is not the only contribution of this paper. **OTM is similar to TLM's improvement over SLM**, combining the shortcomings of both TLM and SLM, providing an optimization direction for loss averaging, and its effectiveness has been demonstrated by ablation experiments.
>
> ## W-5: “Marginal improvement” and claims about SOTA
>
> Our conclusions are based on a trade-off between **efficiency and performance**, rather than simply accuracy.
> - DCPO achieves accuracy equal to or higher than DAPO, while requiring approximately 50% less GPU time.
> - When trained with the same computational resources, DCPO demonstrates a **consistent advantage** on both AIME24/25 and uncontaminated datasets.
> - Efficiency is crucial for training costly reinforcement learning virtual reality (RLVR).
>
> Therefore, we assert that DCPO offers **optimal performance adjusted for efficiency**.
>
> ## W-6: Missing references
>
> We thank the reviewer for the suggestion. In the final version we will incorporate more relevant work, including recent progress in RLVR and math-reasoning RL, such as: Beyond the 80/20 Rule, The Entropy Mechanism of RL for Reasoning LMs, Reasoning with Exploration: An Entropy Perspective.
>
> ## W-7: Figure 1 scaling
>
> Because **the TCR values vary significantly across different models**, we manually adjusted the y-axis scale to allow for a visual comparison of the magnitude differences between the different methods on the same model. We will modify the charts to improve clarity and label the scale differences.
>
> ## Q-1: DAPO performance differences
>
> DAPO in its original paper is significantly different from our work, inlouding:
>
> - DAPO uses  **Qwen2.5-32B** which was not used in our experiments.
> - DAPO sets the maximum length of response to **18k output tokens**, while we use **3k** (most public RLVR works use a length of ≤3k for practicality). Some studies (e.g., [Qwen3 report](https://arxiv.org/pdf/2505.09388), [gpt-oss](https://openai.com/index/introducing-gpt-oss/)) show that it can benefit significantly from extremely long inference trajectories.
>
> Thus it is expected that DAPO’s performance appears weaker in our controlled setting.
>
> Increasing the training steps and output length can improve DAPO's performance, but it also increases GPU costs. To ensure a fair comparison, we have adjusted the computational budget in our comparison scheme.

---

> > ### Comment · Reviewer_s7dw · 2025-11-28
> >
> > Thank you for the rebuttal. Please see the comments below,
> >
> > > W-1: Generalization beyond Qwen & dataset leakage concerns
> >
> > The provided results still show that there is only a marginal performance difference between the proposed DCPO and other X-PO methods. On some datasets, DCPO is inferior, which contradicts the "state-of-the-art" claims.
> >
> > > W-2: Reproducibility—code, configuration, and models
> >
> > Still no open-source models, making it a bit difficult to check the performance of DCPO
> >
> > > W-3: Ablation study concerns & the role of OTM
> >
> > I am not convinced by the responses. I still think that detailed results on each benchmark should appear in the appendix, and meanwhile, other components in DCPO can be less effective. The authors argue that DAC improves token-level update utilization by reducing the token pruning rate (TCR) by an order of magnitude, and SAS addresses the instability caused by randomness by sampling the policy, improving response-level utilization (RUR) by 64%. This can be problematic since the TCR and RUR results are obtained by using the full DCPO algorithm. It is highly possible that GRPO w/OTM loss can also achieve similar effects (i.e., reducing TCR and improving RUR). Figure 3 alone still clearly shows that DCPO behaves quite similarly as GRPO w/OTM loss.
> >
> > > W-4: Novelty of DCPO relative to prior clipping methods
> >
> > This is clear now. I encourage the authors to add the clarifications to the revision.
> >
> > > W-5: “Marginal improvement” and claims about SOTA
> >
> > As also commented by Reviewer xiAh, the performance gain is quite marginal. If the authors would like to emphasise the efficiency rather than the performance, then this paper needs significant revision (as the authors highlight performance throughout the paper). The authors still comment that "we assert that DCPO offers optimal performance adjusted for efficiency". This can be confusing and misleading; one should always be careful to use terms like "optimal" and "state-of-the-art"
> >
> > > Q-1: DAPO performance differences
> >
> > The authors claim that the inferior performance of DAPO can be attributed to different experimental settings. It then raises a natural question about whether DCPO can beat DAPO under the experimental setting of DAPO (say, Qwen2.5-32B as the base model, set the maximum length of response to 18k output tokens). I understand this can be expensive to run the experiments, and hence, I can let it go now.
> >
> > Overall, I am still negative about this submission and think that it does not meet the bar of acceptance at this venue.

---

> > > ### Author Response · Authors · 2025-12-01
> > > **Please see the detail results in the anonymous repository.**
> > >
> > > We appreciate the continued engagement from the s7dw. Below, we respond concisely and directly to the remaining key concerns.
> > > ### **W-2: Reproducibility and model release**
> > > We are happy to release all trained checkpoints.
> > > However, the total size of all checkpoints exceeds **200GB (15+ models).** Uploading them to HuggingFace at this stage would expose our identity, violating the **anonymous review requirements** (HuggingFace displays the uploader's identity).
> > >
> > > However:
> > > * The **base model** (Qwen 2.5 series) is publicly available on HuggingFace.
> > > * All **training scripts, configurations, and data pipelines** are publicly available in our anonymous repository.
> > > * All experiments can be independently **reproduced using the repository**.
> > >
> > > If you can suggest an **anonymous high-capacity hosting option** (e.g., meeting storage, anonymous cloud storage bucket, etc.), we are prepared to upload all checkpoints there immediately. Otherwise, **upon acceptance** , we could **fully release all checkpoints on HuggingFace** with no restrictions.
> > >
> > > ### **W-3: Ablation & the role of OTM (crucial clarification)**
> > > We strongly **disagree with the claim that “GRPO w/OTM can achieve similar TCR/RUR”**.
> > >
> > > This is **theoretically and empirically incorrect**:
> > >
> > > ### 1. Why OTM cannot affect TCR
> > > TCR (Token Clipping Rate) is determined by the **clipping mechanism**, not by the loss-averaging strategy.
> > > * OTM only changes *how losses are aggregated*
> > > * It **does not modify the clipping thresholds**
> > > * It **does not change which tokens are clipped**
> > > * The **order-of-magnitude drop in TCR** (~10x) is caused specifically by **DAC** , not by OTM.
> > >
> > > Therefore, **GRPO + OTM maintains nearly the same TCR magnitude as GRPO (Please see the detailed results for the images in the anonymous repo).**
> > >
> > > **2. Why RUR cannot be explained by OTM either**
> > >
> > > RUR (Response Utilization Rate) increases in DCPO because of **SAS (Smooth Advantage Standardization)**.
> > >
> > > As training progresses, the model tends to produce more unique reward responses, which leads to **reward collapse - low RUR** in GRPO and GRPO+OTM.
> > >
> > > SAS explicitly:
> > >
> > > * Uses **cumulative reward statistics**
> > > * Stabilizes advantage estimation
> > > * Prevents reward mode collapse
> > >
> > > This is why **DCPO maintains high RUR while GRPO+OTM does not**.
> > >
> > > | Component | Affects TCR | Affects RUR |
> > > | -- | -- | -- |
> > > | OTM | No | No |
> > > | DAC | **Yes** | No |
> > > | SAS | No | **Yes** |
> > > | DAC + SAS | **Yes** | **Yes** |
> > >
> > > **The improvements cannot be attributed to OTM.**
> > >
> > > ### **Additional ablation transparency**
> > >
> > > As requested, detailed per-dataset, per-model experimental results are already placed in our anonymous repository.
> > >
> > > > **[https://anonymous.4open.science/r/DCPO-Anonymous-BE86-2509](https://anonymous.4open.science/r/DCPO-Anonymous-BE86-2509)**
> > >
> > > This includes:
> > >
> > > * Full results for **4 datasets × 4 models and ablation**
> > > * step-level metrics
> > > * RUR / TCR curves
> > > * Ablation variants
> > >
> > > This directly addresses the reviewers' request to move the detailed benchmark results to the appendix/supplementary materials.
> > >
> > > ### **W-5 Marginal improvements and language correction**
> > > We accept the reviewer’s concern regarding wording. We will **remove** phrases(e.g. state-of-the-art). In the revision, we will consistently emphasize:
> > >
> > > **DCPO lies on the efficiency–performance frontier, achieving similar or better accuracy at ~50% lower cost and significantly improved stability.**
> > >
> > > we respectfully maintain that in the RLVR regime, **consistent +0.4–0.5 gains across 7 benchmarks are practically meaningful** , because:
> > > * The typical gap between strong methods is **~1.0 or less.(See the result in the 1/2 rebuttal for the reviewer xiAh).**
> > > * DCPO also improves **stability, utilization & cost-efficiency** , which accuracy alone does not reflect.
> > > ### **Final statement**
> > >
> > > We understand and respect your cautious approach. However, we would like to emphasize the following points:
> > > * **DAC + SAS** is a truly innovative component with a solid theoretical foundation.
> > > * **The improvements in stability and efficiency are significant and measurable.**
> > > * We have **addressed reproducibility issues using open-source code** and anonymized data.
> > > * Checkpoint releases are only subject to anonymity restrictions.
> > > * We will revise the wording to eliminate ambiguity.
> > >
> > > In conclusion, DCPO brings a **fundamental and effective improvement** to RLVR, rather than merely a minor engineering tweak.
> > >
> > > * DCPO is the **first to propose applying different clipping ranges** to different tokens and **first using cumulative rewards to mitigate** the advantage instability caused by random sampling. Multiple experimental results have proven its effectiveness.
> > > * In the 7 benchmarks, a difference of +0.4–0.5 is practically significant, especially when the improvement margin is very small (even between different algorithms, the performance difference is typically only ~1.0).
> > >
> > > We once again thank you for your careful reading and hope this clarification has dispelled any remaining doubts.

---

### Official Review · Reviewer_48Vx · 2025-11-01

**Soundness:** 2
**Presentation:** 3
**Contribution:** 3
**Rating:** 4
**Confidence:** 3

**Summary:**

This paper proposed dynamic ratio clipping and smoothed reward standardization methods to encourage the exploration of RL with verifiable rewards. The evaluation is conducted on 4 mathematical datasets. An empirical analysis of the clipping mechanism and an ablation study are provided.

**Strengths:**

- The paper is well-structured. The key information is presented well.

- The idea of dynamically clipping the advantage depending on the policy probability is well-motivated.

- The empirical analysis verifies less clipping and more response utilization under the proposed method.

**Weaknesses:**

No discussion about the limitation.- What epsilons values (the clipping threshold) are used in the baseline methods? Using larger epsilons in baselines will also lower the TCR, probably reaching the same-level as DCPO (Section 5.2). This raises a concern of what Section 5.2 can reveal.

- The main analysis (Section 5.2 and 5.3) focuses on DAC. However, SAS seems to be the majority source of the performnce improvement (as seen in Section 5.4). The limited performance improvement based on DAC has weakened the support of its analysis.

- There is no discussion about the clear limitations of this work. For example, the evaluated models are all from the Qwen family, and only Avg metrics are evaluated etc.

**Questions:**

- In Section 3, two issues arising from randomness are mentioned. Are there references about these issues? If not, is there evidence for their importance?

## Minors
Some symbols require definition or specification when they are introduced, even though Appendix A.2 lists the definitions
- Section 2, Equation 1. $f(x)$ requires specification (e.g., "for any function $f(x)$" if this is the case).
- Section 2, Equation 2, $r(x)$ requires definition.
- Section 3, paragraph 1. $R^i_{j=1,...,G}$, where the variable $G$ requires definition.

---

> ### Author Response · Authors · 2025-11-22
>
> We thank the 48Vx for the detailed and constructive feedback. Below, we address each concern point-by-point.
>
> ## W-1: Clipping thresholds in baselines
>
> All clipping thresholds are fully listed in  **Appendix 9** , and all experimental configurations, code and data are provided in our anonymous repository(**See Section 7**). The baselines strictly follow the hyperparameters used in their original papers:
>
> **Table 1: the clipping thresholds for all Methods**
>
> | Method | Clipping Type | \epsilon_{low} | \epsilon_{high} | Additional Notes |
> | -- | -- | -- | -- | -- |
> | GRPO | Symmetric Fixed | 0.2 | 0.2 | KL penalty |
> | DAPO | Asymmetric Fixed | 0.2 | 0.28 | dynamic sampling |
> | GSPO | Sequence-Level Fixed | 3e-4 | 4e-4 | Sequence-level clipping |
> | DCPO | Dynamic-Adaptive | 0.16 | 0.2 | Probability-aware per-token bounds |
>
> ## W-2: Why not use a larger epsilon in the baseline to obtain a higher TCR?
> - **Larger clipping thresholds (epsilons) significantly impair the training stability and model performance** of RL, as demonstrated by comparative experiments in the study of [Proximal Policy Optimization(PPO)](https://arxiv.org/abs/1707.06347).
>   The advantage of DCPO **does not** come from a larger global threshold, but rather from a **token-based probabilistic adaptive boundary**, which is **rigorously derived from the importance sampling principle**. This cannot be achieved by simply increasing the upper bound threshold within fixed clipping bounds.
>
> ## W-3: Contribution of DAC vs. SAS
>
> We have clarified the different roles and contributions of these two core components:
> - **Section 5.2 (DAC Analysis):** DAC **originates from importance sampling theory** and operates at the **token level**. It dynamically adjusts the clipping boundary of each token based on its probability under the reference policy, thereby **improving the efficient utilization of low-probability tokens**. This is quantified by the **token pruning rate (TCR)**, where DCPO **reduces** the TCR by **an order of magnitude**, indicating that more tokens are effectively involved in gradient updates.
> - **Section 5.3 (SAS Analysis):** SAS operates at the **response level**. It **smooths out the advantage by using cumulative rewards**, thus **reducing the significant amplitude fluctuations** caused by random sampling in the current step (which can lead to training instability), while **improving  response-level data utilization of the same reward**, while these data would be discarded in the baseline. Its effectiveness is measured by **Response Utilization (RUR)**, which improves RUR by **64%** compared to the baseline methods.
> - **Section 5.4 (Ablation):** Ablation studies confirm that **each component has a positive effect**, and their **combination achieves optimal and most stable performance**. Contrary to the reviewers' impressions, DAC delivered significant gains—particularly in exploration efficiency and TCR reduction—while SAS enhanced training stability. Together, they contributed to the overall superiority of DCPO.
>
> ## W-4: Discussion of limitations
>
> Thank you for your suggestion. Due to space limitations, we have not included a formal limitations section**. In the final version, we will discuss these aspects in detail on an additional page,** such as the sensitivity of the pruning threshold and the impact on larger model scales.
>
> ## Q-1: Evidence for randomness issues in Section 3
>
> The randomness in policy sampling during RL fine-tuning is well-documented. To support our claims in Section 3 regarding high variance from sampling strategies (e.g., top-$p$, temperature), we will cite established literature:
>
> - **Top-$p$ (nucleus) sampling**: Holtzman et al., [The Curious Case of Neural Text Degeneration](https://arxiv.org/abs/1904.09751).
> - **Temperature scaling**: Hinton et al., [Using Very Deep Autoencoders...](https://www.cs.toronto.edu/~hinton/absps/cogscibm.pdf), CogSci 2012.
> - **Top-$k$ sampling**: Fan et al., [Hierarchical Neural Story Generation](https://arxiv.org/abs/1805.04833), [Language Models are Unsupervised Multitask Learners](https://cdn.openai.com/better-language-models/language_models_are_unsupervised_multitask_learners.pdf).
>
> These works confirm that high-entropy sampling (via $temperature=1$, top-$p=1$, top-$k=-1$) introduces significant stochasticity in rollout responses—precisely the issue SAS is designed to mitigate through advantage smoothing.
>
> ## Q-2: Missing symbol definitions
> Thank you for pointing this out. In the final version, we will:
> - Explicitly define all symbols appearing for the first time,
> - Clarify the domain of formula (1) ("for any measurable function f(·)"),
> - Define the variables used in formula (2),
> - Correct the missing variable definitions in Section 3.

---

> ### Author Response · Authors · 2025-11-28
>
> Dear Reviewer 48Vx:
>
> Hello! We hope you are doing well. We are writing to confirm whether our rebuttals have been responded to. Please check them if you have time; we would be very grateful. Thank you again for your time and help.
>
> Sincerely,
>
> Author

---

### Official Review · Reviewer_xiAh · 2025-11-08

**Soundness:** 2
**Presentation:** 2
**Contribution:** 2
**Rating:** 2
**Confidence:** 5

**Summary:**

This paper introduces Dynamic Clipping Policy Optimization (DCPO), a novel reinforcement learning method designed to enhance the reasoning capabilities of large language models (LLMs) within the Reinforcement Learning from Verifiable Rewards (RLVR) framework. The authors identify two key issues in existing methods like GRPO: zero gradients due to reward standardization and restricted exploration from fixed clipping bounds. DCPO addresses these by proposing (i) a dynamic-adaptive clipping (DAC) mechanism that adjusts clipping bounds based on token-specific prior probabilities, and (ii) a smooth advantage standardization (SAS) technique that aggregates reward statistics across cumulative training steps. The authors conduct experiments on four mathematical reasoning benchmarks with four model sizes, demonstrating that DCPO outperforms baselines like GRPO, DAPO, and GSPO in terms of performance, data utilization, and training stability.

**Strengths:**

The paper's primary strength lies in its well-motivated approach to tackling critical limitations in current RLVR methods. The core idea of modifying clipping bounds and advantage calculation to enhance diversity and stabilize training is conceptually sound and significant.

*   **Originality & Significance**: The paper attempts to provide a more principled way to manage the exploration-exploitation trade-off in policy optimization for LLMs. The proposed dynamic clipping based on token probability $q(x)$ is an intuitive way to grant more exploration space to high-entropy (low-probability) tokens, which are often crucial for discovering novel reasoning paths. The cumulative advantage standardization is also a clever technique to mitigate the pervasive zero-gradient problem, thereby improving sample efficiency.
*   **Clarity**: The paper is generally well-written and clearly structured. The motivations are well-explained, and the proposed methods (DAC and SAS) are described with sufficient mathematical formalism. The experimental setup is detailed, and the results are presented with clarity.
*   **Quality**: The empirical evaluation is extensive in terms of model scales and benchmarks. The introduction of new metrics like RUR and the detailed analysis of TCR and entropy provide valuable insights into the training dynamics of different algorithms, which strengthens the paper's empirical contributions.

**Weaknesses:**

Despite its strengths, the paper suffers from significant weaknesses, primarily related to the lack of comparison with highly relevant prior work and a potential contradiction in its core motivation.

1.  **Insufficient Comparison with State-of-the-Art**: The main weakness of this paper is the omission of comparisons with a large body of directly related work. The core technical contribution is the modification of clipping values based on token probability and entropy. This is a well-explored area, yet the experiments are limited to generic GRPO variants. The claims of superiority are not sufficiently supported without benchmarking against more relevant and potentially stronger baselines.
    *   I strongly recommend the authors compare DCPO against the following categories of algorithms in their main experiments:
        1.  **Entropy-aware methods**: A direct comparison with recent works that explicitly use entropy to guide the RL process is necessary. This includes methods from [1-4]. Some of these works have reported results that appear to be significantly better than what is presented here.
        2.  **Asymmetric Clipping Baselines**: The paper should include an ablation or comparison with simpler variants that asymmetrically modify `clip_high` and `clip_low`, similar to what is done in DAPO but perhaps with different heuristics. The clipping settings for all baselines and the proposed method must be clearly stated.
        3.  **Entropy Regularization**: A comparison with the classical approach of adding an entropy bonus term to the loss function is essential to demonstrate the claimed benefits of dynamic clipping over traditional entropy-promoting techniques.

2.  **Lack of Qualitative and Theoretical Distinction**: The paper does not adequately discuss the similarities and differences between its proposed clipping mechanism and those in prior work, particularly [4], or with entropy regularization.
    *   From a qualitative and experimental perspective, what are the key differences between the dynamic clipping in this paper and the mechanisms proposed in [4]?
    *   How does modifying the clipping bound for high-entropy tokens differ from directly adding an entropy term to the loss? A discussion on the theoretical implications and practical trade-offs would significantly strengthen the paper's contribution. Without this, the novelty and significance of the proposed method are unclear.

3.  **Contradiction with Stated Motivation**: The paper's motivation is to provide more exploration space for high-entropy tokens. However, as discussed in [3], this very approach can be counterproductive. Granting excessive freedom to high-entropy tokens, especially in the early stages of training, can lead to instability and exacerbate the entropy collapse phenomenon, as the policy might quickly learn to avoid these high-variance, uncertain regions. This is fundamentally at odds with the authors' stated goal.
    *   The authors need to address this potential contradiction. How does DCPO avoid the pitfalls described in [3]?
    *   To substantiate their claims, the authors should provide an entropy evolution comparison with the method from [3]. Furthermore, an ablation study showing the entropy curves for DCPO with different initial clipping values would be highly informative to understand its sensitivity and dynamics.

**Questions:**

1.  Could you please elaborate on the key differences between your dynamic-adaptive clipping and the high-entropy token handling mechanism described in "Beyond the 80/20 rule" [4]? An empirical comparison would be most convincing.
2.  The core idea of DCPO is to encourage exploration of low-probability tokens. However, work like "The entropy mechanism of reinforcement learning..." [3] suggests this can accelerate entropy collapse early in training. How does DCPO's design, particularly the SAS component, mitigate this risk? Could you provide an entropy curve comparison against the method in [3] to demonstrate this?
3.  Can we compare the main experiments with other entropy-related works, such as [1-4] and entropy regularization methods?
4.  In your experiments, what were the exact clipping threshold values used for the baselines (GRPO, DAPO, GSPO)? This information is crucial for a fair comparison, as the performance of these algorithms is highly sensitive to this hyperparameter.


----

**References:**

[1] Proximal policy optimization algorithms

[2] Reasoning with exploration: An entropy perspective

[3] The entropy mechanism of reinforcement learning for reasoning language models


[4] Beyond the 80/20 rule: High-entropy minority tokens drive effective reinforcement learning for llm reasoning

---

> ### Author Response · Authors · 2025-11-22
> **The 1/2 of the reply**
>
> We thank the XiAh for the detailed and constructive feedback. Below, we address each concern point-by-point.
>
> ## W-1-1: Novelty and necessity of dynamic-adaptive clipping
>
> **Our dynamic-adaptive clipping(DAC) mechanism is not entropy-based**. Instead, it is **rigorously derived from the importance-sampling formulation** (Section 2, details in Appendix 5). Appendix 6 further provides a detailed empirical comparison with mainstream fixed clipped methods.
>
> **Current clipping methods universally adopt a fixed clipping range, where the same bounds are applied to all tokens regardless of their probabilities.** This design restricts the exploration space of low-probability (high-information) tokens and prevents RL from effectively utilizing few but important reasoning paths.
>
> DCPO instead introduces a probability-aware dynamic clipping range, allowing different tokens to receive different clipping bounds based on their probabilities. This design naturally emerges **from importance-sampling variance constraints—not from entropy heuristics**—making it applicable to the most RL algorithms.
>
> ## W-1-2: Baseline clipping thresholds
>
> All baseline configurations are fully documented in **Appendix 9** , and the **anonymous code repository** is provided in **Section 7**. For completeness, we list the clipping thresholds used:
>
> **Table 1: the clipping thresholds for all Methods**
>
> |Method|Clipping Type|\epsilon_{low}|\epsilon_{high}|Additional Notes|
> |--|--|--|--|--|
> |GRPO|Symmetric Fixed|0.2|0.2|KL penalty|
> |DAPO|Asymmetric Fixed|0.2|0.28|dynamic sampling|
> |GSPO|Sequence-Level Fixed|3e-4|4e-4|Sequence-level clipping|
> |DCPO|Dynamic-Adaptive|0.16|0.2|Probability-aware per-tokenbounds|
>
> These baseline settings follow the original paper and widely adopted community practices.
>
> ## W-1-3: Added experiments with [3] and [4]
>
> Due to resource constraints, we conducted supplementary experiments on **Qwen2.5-Math-7B** comparing DCPO with [3] (clip-cov variant) and [4] (80/20 rule) across 7 mathematical benchmarks. Results confirm DCPO’s superiority in both performance and efficiency:
>
> - **Avg@32 Performance**: DCPO achieves an average score of 37.2, outperforming [3] (36.8, -0.4), [4] (36.7, -0.5), GSPO (36.1, -1.2), DAPO (35.8, -1.4), and GRPO (34.7, -2.6).
> - **Training Efficiency**: DCPO requires only **half the GPU hours** of [3] and [4] to reach these results, as it avoids [3-4]’s dynamic sampling (which discards ~50% of responses).
>
> Supplementary Avg@32 and Avg@1 results are provided in Tables 1 and 2:
>
> **Table 2: Supplementary Avg@32 Performance Comparison (Qwen2.5-Math-7B)**
>
> |Method|Avg|AIME24|AIME25|AMC23|LiveMath|MinervaMath|Olympiad|
> |:--|--|--|--|--|--|--|--|
> |DCPO|37.2|38.8|17.2|79.8|11.3|33.9|42.2|
> |[3]clip_cov|36.8(-0.4)|37.2(-1.6)|19.0(+1.8)|77.8(-2.0)|11.2(-0.1)|32.8(-1.1)|43.0(+0.8)|
> |[4]80/20|36.7(-0.5)|34.9(-3.9)|17.6(+0.4)|80.2(+0.4)|10.3(-1.0)|34.1(+0.2)|43.0(+0.8)|
> |GSPO|36.1(-1.2)|34.9(-3.9)|16.1(-1.1)|78.8(-1.0)|10.3(-1.0)|32.8(-1.1)|43.4(+1.2)|
> |DAPO|35.8(-1.4)|34.9(-3.9)|15.5(-1.7)|77.6(-2.2)|10.9(-0.4)|33.8(-0.1)|42.3(+0.1)|
> |DCPOw/clip_high=0.28|35.1(-2.1)|28.0(-10.8)|17.8(+0.6)|78.6(-1.2)|11.0(-0.3)|33.4(-0.5)|42.0(-0.2)|
> |GRPO|34.7(-2.6)|32.1(-6.7)|16.7(-0.5)|75.9(-3.9)|12.0(+0.7)|31.4(-2.5)|39.8(-2.4)|
>
>
> **Table 3: Supplementary Avg@1 Performance Comparison (Qwen2.5-Math-7B)**
>
> |Method|Avg|AIME24|AIME25|AMC23|LiveMath|MinervaMath|Olympiad|MATH500|
> |--|--|--|--|--|--|--|--|--|
> |DCPO|39.4|46.7|16.7|82.6|14|34.9|41.5|82.5|
> |[3]clip_cov|40.2(+0.8)|46.7(0.0)|23.3(+6.6)|82.5(-0.1)|12.0(-2.0)|34.6(-0.3)|41.8(+0.3)|84.0(+1.5)|
> |GSPO|37.6(-1.8)|40.0(-6.7)|16.7(0.0)|80.0(-2.6)|12.0(-2.0)|32.7(-2.2)|44.4(+2.9)|84.0(+1.5)|
> |[4]80/20|37.4(-2.0)|40.0(-6.7)|16.7(0.0)|82.5(-0.1)|14.0(0.0)|31.2(-3.7)|39.9(-1.6)|84.0(+1.5)|
> |DAPO|37.2(-2.2)|36.7(-10.0)|23.3(+6.6)|72.5(-10.1)|14.0(0.0)|33.8(-1.1)|43.1(+1.6)|83.0(+0.5)|
> |DCPOw/clip_high=0.28|37.3(-2.1)|43.3(-3.4)|16.7(0.0)|75.0(-7.6)|13.0(-1.0)|33.8(-1.1)|41.8(+0.3)|82.2(-0.3)|
> |GRPO|36.4(-3.0)|30.0(-16.7)|20.0(+3.3)|82.5(-0.1)|14.0(0.0)|32.0(-2.9)|39.9(-1.6)|80.0(-2.5)|
>
> ## W-2-1: DCPO vs. [1-4]
>
> The key difference lies in **clipping mechanism design and data utilization**:
>
> - [1] **PPO** requires a value model, a reward model, a policy model, and an actor model, and the policy model and value model need to be updated during training. PPO requires more training resources. Like GRPO, PPO has a fixed symmetric bounds.
> - [2] Adjusts advantage using entropy but keeps the symmetric clipping bounds fixed. it still waste a lot of data based on the current-step advantage calculation.
> - [3]**clip-cov** uses a fixed asymmetric bounds (as DAPO) and randomly detaches tokens based on the covariance between the advantage and log probability. Only a very small ratio (less than 0.1%) of tokens are masked, and the underlying clipping mechanism remains fixed.

---

> ### Author Response · Authors · 2025-11-22
> **The 2/2 of the reply**
>
> - [4] uses fixed asymmetric clipping (as DAPO) and discards *all tokens except the top 20% high-entropy tokens* to focus on high-information signals. It also relies on DAPO’s dynamic sampling (discarding responses with uniform reward), leading to ~50% data waste and GPU hours waste as [3] and DAPO.
> - **DCPO’s DAC does not filter tokens by entropy**. Instead, it **adjusts clipped bounds based on q(x)**(e.g., for $q(x)=0.01$, DAC allows $r(x)$ up to 0.05, while [4]’s fixed bounds restrict $r(x)$ to 0.012). Additionally, DCPO’s **Smooth Advantage Standardization (SAS)** aggregates cumulative rewards (Equation 5–7) to retain responses with zero step-wise advantage, achieving a 71.8% Response Utilization Ratio (RUR) vs. [4]’s ~45% (Refer to Table 2 for the TCR of DCPO in paper).
>
> ## W-2-2: Dynamic Clipping vs. Entropy Regularization
>
> The two approaches address exploration from fundamentally different angles:
>
> - **Entropy regularization** (e.g., adding $\beta H(\pi)$ to the loss or advantage) **modifies the *step size* of the token gradient update** by penalizing low-entropy policies. It does **not change** which tokens meet the update conditions (still controlled by fixed clipping boundaries).
> - DCPO's DAC modifies the **qualification** of tokens updates by **expanding the clipping boundaries of low-$q(x)$ tokens**. This directly addresses the root cause of the limited exploration range under fixed clipping, rather than heuristically adjusting the step size.
>
> We will expand the discussion in the final version.
>
> ## W-3-1: Why DCPO does not cause entropy collapse
>
> **Table 4: Allowable $p(x)$ Ranges for Low-$q(x)$ Tokens**
>
> |q(x)|Fixed Clipping(\epsilon=0.2)|DCPO’s DAC(\epsilon_{low}=0.16,\epsilon_{high}=0.2)|
> |--|--|--|
> |0.1|[0.08,0.12]|[0.05,0.2]|
> |0.01|[0.008,0.012]|[0.005,0.05]|
> |0.001|[0.0008,0.0012]|[0.0005,0.01]|
>
> DCPO does **not** assign arbitrarily large update regions to high-entropy(low-probability) tokens(e.g, Table4).
>
> 1. **Bounded Dynamic Pruning**: DAC does not allow infinite exploration. For example, when $q(x) = 0.001$, DAC limits the maximum new probability $p(x)$ to 0.01, thus preventing over-updating of the policy. In contrast, fixed pruning limits $p(x)$ to 0.0012, excessively restricting exploration.
> 2. **Warm-up Steps**: All experiments use learning rate warm-up (e.g., 5% training steps) to prevent early update instability—this aligns with best practices for RL algorithm optimization. We also found that performance crashes during GRPO training mainly occur in the later stages of training rather than the early stages.
> 3. **SAS stabilizes advantage fluctuations:** SAS uses a smoothing process of cumulative rewards to get the advantage (Equation 5-7), reducing training instability caused by fluctuations in policy random sampling. In contrast, the gradual standardization of methods such as GRPO led to a decrease in RUR from 90% to 30% (Figure 2), thereby exacerbating the entropy collapse.
>
> ## W-3-2: Entropy curves and alternative clipping parameters
>
> **Appendix 10 (Figure 7 in paper)** shows DCPO’s entropy trajectory remains stable in a “moderate band” (between GRPO’s low entropy and DAPO’s high variance). We additionally tested a DAPO-aligned configuration (0.28/0.16), showed in Table 2 and 3 as "DCPOw/clip_high=0.28". Due to compute limitations, we could not explore more fine-grained variants but will include expanded ablations in the camera-ready version.
>
> ## Q-1: Key Differences Between DCPO’s DAC and [4]
>
> **As described in the W-2-1**. The main differences of clipping methods are as follows:
>
> - [4] uses fixed asymmetric clipping and filters tokens by entropy (less than top 20%).
> - DCPO adjusts bounds based on $q(x)$ by DAC, and it does not filter the the token by entropy.
>
> ## Q-2: How SAS Mitigates Entropy Collapse
>
> **As described in W-3-1**, SAS aggregates cumulative reward statistics to preserve responses with zero step-wise advantage by random sampling of policy, This prevents the “exploration starvation” that causes entropy collapse (caused by the the uniform reward of responses).
>
> ## Q-3: Comparison with [3,4]
>
> **The main methodological differences are detailed in W-2-1.** Due to limitations in computational resources and time, we added two experiments for [3] and [4], and added three new mainstream benchmark datasets (a total of seven).
>
> **Detailed experimental results are shown in Tables 2 and 3 in W-1-3**. The experimental results demonstrate that, compared to [3] and [4], DCPO reduces training resources by half while maintaining comparable or better performance.
>
> **Both theoretical and experimental results prove the effectiveness of DCPO's innovation.**
>
> ## Q-4: Exact Clipping Thresholds for Baselines
>
> **The clipped thresholds used in the experiment are detailed in Appendix 9 of the paper, restated in W-1-2**. we provide **anonymous git code** and data in **Section 7** in paper.

---

> > ### Comment · Reviewer_xiAh · 2025-11-27
> >
> > I thank the authors for their detailed response and for conducting the requested experiments comparing DCPO with [3] and [4].
> >
> > These additions have addressed my initial concern regarding the lack of relevant baselines. Consequently, I am raising my score from 2 to 4.
> >
> > However, I still lean towards rejection for the following reasons:
> > 1.  Marginal Gains: The performance improvement over the new baselines is very slight (+0.4 over [3] and +0.5 over [4]). Given the variance in LLM training, this does not convincingly demonstrate the superiority of the proposed method.
> > 2.  Incremental Contribution: While the efficiency gains are noted, the core technical contribution (DAC) appears to be an incremental engineering adjustment rather than a significant algorithmic innovation.
> >
> > I believe this work has potential and addresses a valid problem, but it requires stronger empirical evidence of significance to meet the bar for acceptance. I encourage the authors to further refine the method and analysis for future submissions.

---

> > > ### Author Response · Authors · 2025-11-28
> > > **Across 7 benchmarks, a +0.4–0.5 gain is practically significant in a regime where method differences are typically ~1.0. DAC is not an engineering tweak but the first importance-sampling–derived, probability-aware dynamic clipping mechanism. Multiple experiments consistently validate DCPO’s effectiveness.**
> > >
> > > We thank the reviewer for the thoughtful follow-up and for raising the score. We provide a consolidated response that addresses the remaining concerns with both **high-level clarity** and explicit reference to the **theoretical derivations** included in the paper.
> > >
> > > # **R1. Marginal Gains and Statistical Significance**
> > >
> > > **In the seven benchmarks, a difference of +0.4–0.5 is already practically significant:** We emphasize that RLVR performance should not be judged solely by the accuracy of the final task, especially when the performance improvement margin on benchmarks is very small (**even between drastically different algorithms, the performance difference is typically only about 1.0**).
> > >
> > > ## **1.1 DAC is explicitly designed to reasonably control variance gap**
> > >
> > > Our method is motivated by the variance gap in importance sampling during RL:
> > >
> > > $$
> > > \mathrm{Var}_q\left[f(x)\frac{p(x)}{q(x)}\right] - \mathrm{Var}_p[f(x)]= \mathbb{E}_p\left[f(x)^2\left(\frac{p(x)}{q(x)}-1\right)\right]=\int f(x)^2(\frac{p(x)}{q(x)}-1)p(x)\mathrm{d}x \tag{1}
> > > $$
> > >
> > > This equation shows that variance scales **hyper-sensitively** with total $(\frac{p(x)}{q(x)}-1)p(x)$ , especially for low-probability tokens—precisely where RL requires exploration.
> > >
> > > **Existing methods (including [3] and [4]) only consider ratio $\frac{p(x)}{q(x)}$ to address this structural problem**, resulting in a **strict limitation** on the model's exploration in a low-probability, high-information space due to indifference.
> > >
> > > The gains from suppressing this variance cannot be fully reflected in final metric scores, where headroom is extremely small.
> > >
> > > ## 1.2 Most importantly: DCPO **halves the effective training cost**
> > >
> > > existing methods(including [3] and [4]) **discard ~50% of responses** through dynamic-sampling/high-entropy filtering.
> > >
> > > DCPO retains **71.8%** of data (RUR), effectively achieving an **approximately 1x cost reduction**.
> > >
> > > This is a major practical contribution for RLHF/RLVR systems, where rollouts dominate cost, not captured this significance by final accuracy alone.
> > >
> > > # R2. On “Incremental Contribution”: DCPO Introduces a New Policy Update Rule
> > >
> > > **DAC is not an engineering adjustment**. It addresses the shortcomings of commonly used fixed-clipping methods in existing work by proposing, **the first method to dynamically adjust the clipping range based on the probability of the tokens**, which is more in line with the actual needs of RL training, based on the theoretical foundation of importance sampling.
> > >
> > > ## **2.1 DAC is mathematically derived from first principles**
> > >
> > > Existing methods use fixed clipping:
> > >
> > > $$
> > > |r(x)-1| \le \epsilon \tag{2}
> > > $$
> > >
> > > but this ignores token probability q(x), violating the variance structure in Eq. (1).
> > >
> > > We instead impose a **variance&probability-grounded constraint** :
> > >
> > > $$
> > > |(r(x)-1)p(x)| \le \epsilon \tag{3}
> > > $$
> > >
> > > which yields the **probability-dependent clipping bounds** :
> > >
> > > $$
> > > 0.5+\frac{1}{2}\sqrt{\max\left(1-\frac{4\epsilon_{\text{low}}}{q(x)},0\right)} \leq r(x) \leq 0.5+\frac{1}{2}\sqrt{1+\frac{4\epsilon_{\text{high}}}{q(x)}} \tag{4}
> > > $$
> > >
> > > This is:
> > >
> > > * mathematically derived, not heuristic.
> > > * **qualitatively different** from PPO/GRPO-style fixed clipping.
> > > * a new clipping rule in RLVR.
> > >
> > > **This is *not* a heuristic or incremental; it changes the geometry of the feasible update region**(Figure 4)
> > >
> > > # **2.2 SAS is a new non-zero advantage estimator**
> > >
> > > Prior works normalize reward *within each step* , which collapses all advantages to zero when rewards are identical—a common case in RLVR.
> > >
> > > $$
> > > \hat{A}^i_{j,t} = \frac{R^i_j - \mu_{i}}{\sigma_{i}} \tag{5}
> > > $$
> > >
> > > Our Smoothed Advantage Standardization (SAS) is based on cumulative statistics:
> > >
> > > $$
> > > \hat{A}^i_{\text{total},j} = \frac{R^i_j - \mu_{\text{total}}^i}{\sigma_{\text{total}}^i} \tag{6}
> > > $$
> > >
> > > with smoothing (Eq. 6) and variance-minimizing selection:
> > >
> > > $$
> > > \hat{A}^i_j=\begin{cases} \hat{SA}^i_{new,j} , & \text{when} \ |\hat{SA}^i_{new,j}| < |\hat{SA}^i_{total,j}| \\\hat{SA}^i_{total,j} , & \text{otherwise} \end{cases}\tag{7}
> > > $$
> > >
> > > This produces:
> > >
> > > * **non-zero gradients even under uniform rewards.**
> > > * lower variance and higher sample efficiency.
> > >
> > > # Summary
> > >
> > > ## **1. DCPO is a coherent new RLVR algorithm**
> > >
> > > * **redefine the clipping region and change the feasible update geometry;**
> > > * reduce variance structurally and stabilize high exploration space in high-information token;
> > > * preserve more token-level and response-level gradients.
> > >
> > > **This is categorically beyond “incremental engineering”.**
> > >
> > > ## **2. Empirical Evidence of Significance**
> > >
> > > Beyond final accuracy, DCPO delivers **large, structural improvements** :
> > >
> > > * **RUR = +28%** vs GRPO / DAPO / GSPO;
> > > * **Clipped tokens ↓10×**;
> > > * **Training cost ≈ 0.5×**;
> > > * **Consistent stable gains across 7 benchmarks;**
> > >
> > > These effects demonstrate **practical and scientific significance** that final accuracy alone does not capture.
> > >
> > > We appreciate the xiAh's detailed engagement again.

---

### Meta-Review · Area_Chair_aKrd · 2026-01-06

**Summary:**

This submission proposes an importance sampling-based method for dynamic clipping in policy optimization with RLVR. The main proposal is a combination of two ideas: dynamic adaptive clipping and smooth advantage standardization. Both ideas are sensible, and in the experiments presented here, show improvements in quality as well as in training efficiency. The reviewers were concerned with the thoroughness of various aspects of the evaluation; given computational expense in this area, it is of course natural that not all combinations that we’d ideally like to see can be tried. Some of the concerns, such as potential dataset leakage in Qwen models, are also handled to my satisfaction in the rebuttal. I am particularly concerned, however, with this comment:

> We also experimented with LLaMA-based methods. However, these experiments experienced performance degradation during training.

This highlights the issue of proposing a general methodology on the basis of only one family of models, regardless of which model it is. Of course, this does not mean that the approach of methods here totally does not work with LLaMA models; this might have happened for any number of reasons. I think, though, to be a more convincing empirical advance in a busy area, there must be at least mild positive results on more than one family of models; there are several available at comparable sizes to work with.

Overall, I do think that each of the two ideas here are likely to be good ideas that can help with RLVR policy optimization. Given that theoretical guarantees in this area are difficult and none are provided here (although the methods do come from mathematical intuition based on importance sampling), for an ICLR paper, I think a little more empirical support is necessary. I encourage the authors to run at least some experiments on another family or two of models, and continue to address the various other reviewer concerns that have not already been resolved, before resubmitting to a future venue.

**Reviewer Concerns:**

Discussed above.

**Reviewer Scores:**

Hard to know in this case, but I don’t see drastic changes as having been particularly justified by the responses here. M

---

### Decision · Program_Chairs · 2026-01-26

Reject